# HarmoMoE: Unifying Domain-Specialized Experts into a Mixture-of-Experts Model under Privacy Constraints

## Abstract

Mixture-of-Experts (MoE) models offer a powerful way to scale capacity, but existing designs typically assume centralized access to all training data. In many real-world scenarios, however, data are distributed across clients from different domains and cannot be shared due to privacy constraints, making it challenging to build a unified and generalizable MoE. We propose HarmoMoE, a framework that unifies domain-specialized experts into a single MoE without sharing private data. HarmoMoE combines relevance-weighted DPP proxy selection with a context-aware router, ensuring that experts trained on both private and proxy data remain compatible and effectively coordinated. Experiments on CV and NLP show that HarmoMoE consistently outperforms recent methods such as BTX and FlexOlmo.

## 1 Introduction

Large foundation models [26, 27, 30, 31] have become indispensable across domains such as computer vision (CV) and natural language processing (NLP). In practice, organizations and users often finetune a shared seed model on their own private data, resulting in a collection of specialized experts. Unifying these experts by aggregating their private data to train a single, powerful model (e.g., a Mixture-of-Experts (MoE) [6, 15]) promises broader capabilities. However, data sharing is often infeasible due to confidentiality, regulatory, and ethical constraints. This raises a fundamental question: *How can we merge independently trained experts into one deployable model while strictly preserving data privacy?* Although federated learning [17, 21, 48] enables collaborative training without data sharing, it requires costly synchronized optimization across many clients and often suffers from performance degradation under heterogeneous client data distributions [44, 49].

Several approaches have been proposed to unify independently finetuned experts. Branch-Train-Merge (BTM) [20] ensembles expert predictions, allowing embarrassingly parallel training but failing to produce a single deployable model for downstream fine-tuning or RLHF [29]. Model averaging methods such as Model Soup [45] merge parameters directly, which is computationally efficient but fragile when experts diverge in function space. BTX [39] extends this line by transplanting expert feed-forward network (FFN) sublayers into a shared MoE architecture with a learned router. However, its requirement for client-specific data to train this router limits its use in privacy-sensitive scenarios.

Most recently, FlexOlmo [38] has emerged as the state of the art by anchoring experts to a public model and aligning them through proxy data with router embeddings, thereby avoiding centralized training. However, because each expert is trained only on its own private data, the resulting models are often *domain-isolated*, making it difficult for a router—trained later without access to private data—to coordinate them effectively. Moreover, its reliance on *similarity-based proxy selection* and *shallow router design* further limits expert alignment and contextual specialization. These challenges underscore the need for a more powerful framework that can unify experts under privacy constraints.

We propose **HarmoMoE**, a harmonized MoE framework that unifies diverse experts into a single MoE model while preserving data privacy. Since directly sharing private data is infeasible, our method relies on a carefully selected set from public data that serve as a proxy to approximate private data. To select proxies, we propose a *relevance-weighted determinantal point process (DPP)*, a novel extension of the standard DPP [18, 24], which augments the classical diversity-promoting mechanism with client-specific relevance scores, ensuring that proxies are not only diverse but also relevant to the

client domain. Each client then performs *proxy-aligned expert training*, fine-tuning the FFN sublayers on its private data alongside the proxy samples. Since the same proxies are later used for router training, exposing experts to them during fine-tuning ensures compatibility with the supervision available at unifying time, mitigating domain isolation and enabling effective coordination across experts. We further design a *context-aware router* that combines token-level and sequence-level context for accurate expert assignment, and finally merge all experts' FFN sublayers into MoE layers, finetuned jointly on the unioned proxy data. Experiments on both CV and NLP benchmarks demonstrate that HarmoMoE consistently outperforms recent state-of-the-art baselines.

Our contributions are summarized as follows: (i) We propose HarmoMoE, a privacy-preserving framework that unifies independently finetuned experts into a single MoE model without sharing private data. (ii) We introduce proxy-aligned expert training with a relevance-weighted DPP proxy selection strategy, ensuring effective expert alignment and overcoming the redundancy of similarity-only sampling. (iii) We design a context-aware router and demonstrate, through extensive CV and NLP experiments, that HarmoMoE achieves superior performance over the recent state-of-the-arts.

## 2 RELATED WORKS

### 2.1 FEDERATED LEARNING

Federated Learning (FL) [17, 21, 48] enables collaborative training without centralizing raw data. Classical methods such as FedAvg [25] aggregate local updates, with extensions for parameter-efficient tuning [12] or differential privacy [4]. Yet scaling FL to large models [26, 27, 31] is difficult due to costly synchronization, degraded generalization [49], and privacy leakage from gradients [2, 43]. In contrast, HarmoMoE trains experts independently and asynchronously, then merges them via proxy-data-driven routing, avoiding FL's communication bottlenecks while preserving privacy.

### 2.2 MODEL MERGING AND MIXTURE OF EXPERTS

**Model Merging** [14, 32, 45–47] explores how to combine multiple independently trained models into a single, stronger model without requiring costly joint training. *Branch-Train-Merge (BTM)* [20] trains domain experts independently and ensembles outputs at inference time, but does not yield a single unified model (hindering downstream SFT/RLHF [29] and incurring inference overhead). *Model Soup* [45] shows that averaging the weights of finetuned models often improves both accuracy and robustness with no additional inference cost, yet it is fragile when experts diverge in function space, leading to degraded performance in heterogeneous settings. *Branch-Train-MiX (BTX)* [39] inserts experts into MoE layers and learns a post-hoc router via additional fine-tuning on private data. More recently, *FlexOlmo* [38] advances this direction in federated settings by anchoring experts to a shared public model and aligning them with router embeddings, where proxy samples are selected for each client *based solely on similarity*. In contrast, our HarmoMoE leverages a relevance-and-diversity criterion via a relevance-weighted DPP to select proxies, and employs a context-aware router, thereby harmonizing heterogeneous experts into a unified MoE model under privacy constraints.

The **Mixture of Experts (MoE)** [15] framework enhances model flexibility by combining specialized experts via a gating mechanism. Modern MoE architectures [6, 33] scale Transformers by activating only a sparse subset of experts per token, expanding capacity without proportional compute cost [37]. Key designs include the Switch Transformer [6] with top-1 routing and variants [1, 33, 34, 37] exploring top-$k$ gating, stochastic routing, and random assignment for better accuracy, efficiency, and load balance. While MoE is effective for scaling, these methods rely on centralized access to all training data and synchronized training. To overcome these limits under privacy constraints, we propose HarmoMoE, which introduces proxy-data-driven routers and decentralized expert training.

### 2.3 DETERMINANTAL POINT PROCESSES (DPPs)

Determinantal Point Processes (DPPs) [18, 19, 24] are probabilistic models designed to capture *negative interactions* among items, which are useful for diverse subset selection. Formally, consider a discrete set $\mathcal{Z} = \{1, 2, \ldots, N\}$, where each element corresponds to an item with feature vector $\mathbf{x}_i$. A DPP defines a probability distribution over all $2^N$ subsets of $\mathcal{Z}$. To specify the distribution, we construct a positive semi-definite kernel matrix $\mathbf{L} \in \mathbb{R}^{N \times N}$, $\quad \mathbf{L}_{ij} = \kappa(\mathbf{x}_i, \mathbf{x}_j)$, where $\kappa(\cdot, \cdot)$ is a

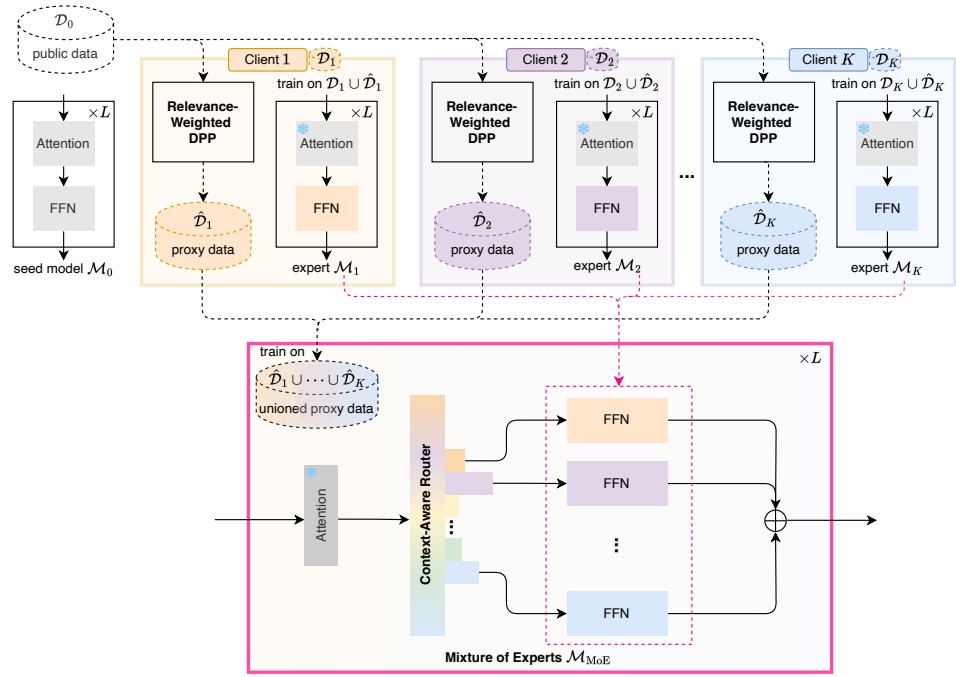

Figure 1: Illustration of HarmoMoE.

kernel function encoding similarity. The probability of sampling a subset $\mathcal{S} \subseteq \mathcal{Z}$ is given by

$$\mathbb{P}(\mathcal{S}) = \frac{\det(\mathbf{L}_{\mathcal{S}})}{\det(\mathbf{L} + \mathbf{I})}, \tag{1}$$

where $\mathbf{L}_{\mathcal{S}}$ is the submatrix indexed by $\mathcal{S}$, $\det(\cdot)$ is the determinant, and $\mathbf{I}$ is the identity matrix. The denominator $\det(\mathbf{L} + \mathbf{I})$ is constant with respect to the choice of $\mathcal{S}$, so for subset selection it can be ignored; maximizing the selection probability $\mathbb{P}(\mathcal{S})$ is thus equivalent to maximizing $\det(\mathbf{L}_{\mathcal{S}})$.

By reproducing kernel Hilbert space representation [36], one may write $\kappa(\mathbf{x}_i, \mathbf{x}_j) = \phi(\mathbf{x}_i)^{\top}\phi(\mathbf{x}_j)$ for some feature map $\phi(\cdot)$. Then $\det(\mathbf{L}_{\mathcal{S}})$ equals the squared volume of parallelotope spanned by $\{\phi(\mathbf{x}_i) \mid i \in \mathcal{S}\}$, so similar items are less likely to be selected together, promoting diversity.

## 3 METHODOLOGY

### 3.1 PROBLEM FORMULATION

Denote by $\mathcal{M}_0$ a seed model and by $\mathcal{D}_0$ a publicly available dataset. We consider $K$ clients, where each client $p$ owns a private dataset $\mathcal{D}_p$ from its local domain. Direct sharing of $\{\mathcal{D}_p\}_{p=1}^K$ is prohibited due to privacy constraints. Each client adapts the seed model to obtain a domain-specialized expert $\mathcal{M}_p$. Our objective is to integrate these experts $\{\mathcal{M}_p\}_{p=1}^K$ into a unified Mixture-of-Experts (MoE) model $\mathcal{M}_{\mathrm{MoE}}$ that serves across all domains. In the MoE architecture, experts encode domain-specific knowledge, while the router coordinates the experts to enable effective collaboration.

The core challenge lies in *training the router*. Conventional MoE training [15, 37] assumes centralized access to all client data, but in our setting only the public dataset $\mathcal{D}_0$ is globally accessible. Thus, the router must be learned without directly observing $\{\mathcal{D}_p\}_{p=1}^K$, while still generalizing across clients' domains. We address this challenge by selecting proxy samples from $\mathcal{D}_0$ to approximate each $\mathcal{D}_p$, enabling the router to coordinate domain-specific experts in a privacy-preserving manner. Figure 1 illustrates our proposed HarmoMoE, consisting of three stages (proxy data selection, proxy-aligned expert training, and context-aware router training) and will be detailed in the following sections.

## 3.2 PROXY DATA SELECTION VIA RELEVANCE-WEIGHTED DPP

Since client private data is inaccessible for router training, we construct a proxy dataset $\hat{\mathcal{D}}_p$ for each client $p$ from the public dataset $\mathcal{D}_0$ to approximate $\mathcal{D}_p$. Effective proxy data should satisfy two criteria: they should be both **relevant** to the private data $\mathcal{D}_p$ and sufficiently **diverse** to avoid redundancy. Relevance ensures that the proxy samples resemble the private data so that the router trained on proxies learns domain-appropriate decision boundaries, rather than being distracted by unrelated public samples. Diversity, on the other hand, ensures that the selected proxy samples cover different regions of the private-data manifold, rather than clustering around a narrow region of highly similar samples, thereby providing broader coverage and improving the router's generalization.

Determinantal point processes (DPPs) [18, 19, 24] naturally enforce diversity, but a vanilla DPP (Section 2.3) ignores whether the chosen samples align with the client's domain $\mathcal{D}_p$. Hence, naively applying a vanilla DPP may select a diverse yet irrelevant proxy dataset. To overcome this, we propose a *relevance-weighted DPP*, which augments the kernel with client-specific relevance scores:

$$\tilde{\kappa}(\mathbf{x}_i, \mathbf{x}_j) = g(\mathbf{x}_i, \mathcal{D}_p)\, \kappa(\mathbf{x}_i, \mathbf{x}_j)\, g(\mathbf{x}_j, \mathcal{D}_p), \tag{2}$$

where $\kappa(\mathbf{x}_i, \mathbf{x}_j)$ measures similarity between public samples (e.g., cosine similarity), and $g(\mathbf{x}_i, \mathcal{D}_p)$ quantifies the relevance of $\mathbf{x}_i$ to $\mathcal{D}_p$ (e.g., via a classifier distinguishing $\mathcal{D}_0$ from $\mathcal{D}_p$, see Appendix A). This yields the *relevance-weighted kernel matrix*

$$\widetilde{\mathbf{L}} = \mathrm{Diag}(\mathbf{r})\, \mathbf{L}\, \mathrm{Diag}(\mathbf{r}), \tag{3}$$

where $\mathbf{L}_{ij} = \kappa(\mathbf{x}_i, \mathbf{x}_j)$, $\mathbf{r} = [g(\mathbf{x}_1, \mathcal{D}_p), \ldots, g(\mathbf{x}_N, \mathcal{D}_p)]$, and $\mathrm{Diag}(\mathbf{r})$ denotes the diagonal matrix with $\mathbf{r}$ on the diagonal. According to (1) and (3), the unnormalized log-probability of selecting a subset $\mathcal{S}$ under relevance-weighted DPP is given by

$$\log \det(\widetilde{\mathbf{L}}_{\mathcal{S}}) = 2 \sum_{i \in \mathcal{S}} \log \mathbf{r}_i \; + \; \log \det(\mathbf{L}_{\mathcal{S}}), \tag{4}$$

where the first term $2 \sum_{i \in \mathcal{S}} \log \mathbf{r}_i$ encourages relevance by favoring samples closer to the client's data $\mathcal{D}_p$, while the second term $\log \det(\mathbf{L}_{\mathcal{S}})$ is the standard DPP repulsion term that enforces diversity.

The proxy dataset is selected as $\hat{\mathcal{D}}_p = \arg\max_{\mathcal{S} \subseteq \mathcal{Z}, |\mathcal{S}| = m} \det(\widetilde{\mathbf{L}}_{\mathcal{S}})$, yielding a *relevance-weighted, diverse cover* of the private-data manifold. Since exact maximum a posteriori (MAP) inference is NP-hard, we adopt efficient greedy algorithms with approximation guarantees [8, 9, 18]. We first restrict a candidate pool (the top-$n$ public samples ranked by $g(\mathbf{x}_i, \mathcal{D}_p)$), and then perform greedy MAP inference to construct $\hat{\mathcal{D}}_p$ by iteratively adding the sample that maximizes the marginal gain in $\log \det(\widetilde{\mathbf{L}}_{\hat{\mathcal{D}}_p})$. Using Cholesky updates [11], the computational cost is reduced from $O(nm^3)$ to $O(nm)$ (see Appendix B), where $n$ is the candidate pool size and $m$ is the target proxy set size.

Compared with FlexOlmo's relevance-only proxy selection [38], which often collapses onto redundant samples and yields a narrow view of a client's domain, relevance-weighted DPP explicitly balances relevance and diversity. The diversity term prevents the router from seeing only near-duplicate proxies, giving HarmoMoE a richer supervision signal that better spans the private-domain manifold and enables more effective expert coordination. Since all proxies are drawn from a public dataset, selecting similar public samples for each client does not violate privacy constraints, see Appendix H.5 for more details.

## 3.3 PROXY-ALIGNED EXPERT TRAINING

For each client $p$, we branch from the seed model $\mathcal{M}_0$ and finetune only the feed-forward network (FFN) sublayers using both its private data $\mathcal{D}_p$ and proxy data $\hat{\mathcal{D}}_p$, while keeping all other layers frozen. If training were performed solely on private data $\mathcal{D}_p$ as used in [38], the resulting experts would indeed become highly specialized to their own domains. However, such specialization is often *domain-isolated*: the router—later trained *without access to private data*—would struggle to coordinate across experts adapted to different domains. Incorporating the *client-specific* proxy data $\hat{\mathcal{D}}_p$ during expert training ensures that the $p$-th expert is calibrated to the data *that is actually available for router learning*. Since the router is trained on the union $\hat{\mathcal{D}}_1 \cup \cdots \cup \hat{\mathcal{D}}_K$, exposing each expert to its own proxy encourages alignment between expert behavior (driven by $\mathcal{D}_p$) and router supervision (driven by proxies), thereby improving routing compatibility while preserving privacy.

---

**Algorithm 1** HarmoMoE.

---

**Require:** public dataset $\mathcal{D}_0$; client private datasets $\{\mathcal{D}_p\}_{p=1}^K$; kernel $\kappa(\cdot, \cdot)$; proxy dataset size $m$; optional candidate pool size $n$ ($m \leq n \ll |\mathcal{D}_0|$);

1: **for** each client $p = 1, 2, \ldots, K$ **do**
2:      // select proxy samples by relevance-weighted DPP
3:      Train a binary classifier $g(\mathbf{x}, \mathcal{D}_p)$ to distinguish $\mathcal{D}_0$ vs. $\mathcal{D}_p$;
4:      Compute relevance $r(\mathbf{x}) = g(\mathbf{x}, \mathcal{D}_p)$ for each $\mathbf{x} \in \mathcal{D}_0$;
5:      Let $\mathcal{C}_p \subseteq \mathcal{D}_0$ be the top-$n$ elements of $\mathcal{D}_0$ ordered by $r(\mathbf{x})$;
6:      Construct kernel matrix $\mathbf{L} \in \mathbb{R}^{n \times n}$ with $\mathbf{L}_{ij} = \kappa(\mathbf{x}_i, \mathbf{x}_j)$ for $\mathbf{x}_i, \mathbf{x}_j \in \mathcal{C}_p$;
7:      Form $\mathbf{r} = [\, r(\mathbf{x}) \,]_{\mathbf{x} \in \mathcal{C}_p}$ and relevance-weighted kernel matrix $\widetilde{\mathbf{L}} = \mathrm{Diag}(\mathbf{r}) \, \mathbf{L} \, \mathrm{Diag}(\mathbf{r})$;
8:      Greedily build $\hat{\mathcal{D}}_p$ of size $m$ by iteratively adding $\mathbf{x}^\star = \arg\max_{\mathbf{x} \in \mathcal{C}_p \backslash \hat{\mathcal{D}}_p} \log \det(\widetilde{\mathbf{L}}_{\hat{\mathcal{D}}_p \cup \{\mathbf{x}\}}) - \log \det(\widetilde{\mathbf{L}}_{\hat{\mathcal{D}}_p})$;
9:      // proxy-aligned expert training
10:     Finetune the FFN sublayers of the $p$-th client model on $\hat{\mathcal{D}}_p \cup \mathcal{D}_p$, and freeze all other sublayers;
11:     For each layer $l$, compute router vector $\mathbf{e}_p^{(l)} = \frac{1}{|\hat{\mathcal{D}}_p \cup \mathcal{D}_p|} \sum_{\mathbf{x} \in \hat{\mathcal{D}}_p \cup \mathcal{D}_p} \mathcal{M}_p^{(1:l)}(\mathbf{x})$;
12: **end for**
13: // build the MoE model with context-aware router
14: Merge all clients' FFN sublayers into a single MoE model $\mathcal{M}_{\mathrm{MoE}}$;
15: Collect $\{\mathbf{e}_p^{(l)}\}_{p=1}^K$ for all layers $l = 1, \ldots, L$ to initialize the router;
16: Compute the routing distribution $\pi^{(l)}(\mathbf{z}_t^{(l)})$ for each layer $l$ using (6);
17: Finetune $\mathcal{M}_{\mathrm{MoE}}$ on the unioned proxy data $\hat{\mathcal{D}}_1 \cup \cdots \cup \hat{\mathcal{D}}_K$;
18: **return** Trained MoE model $\mathcal{M}_{\mathrm{MoE}}$.

---

Unlike FlexOlmo [38], which trains experts only on private data and thus leaves them domain-isolated, our proxy-aligned strategy jointly exposes experts to private and proxy data. This dual alignment preserves domain-specific expertise from $\mathcal{D}_p$ while ensuring compatibility with proxy-based routing, enabling HarmoMoE to achieve more effective expert coordination under privacy constraints.

## 3.4 CONTEXT-AWARE ROUTER FOR EXPERT INTEGRATION

After collecting domain-specific experts $\{\mathcal{M}_p\}_{p=1}^K$, we merge their FFN sublayers into MoE modules at each Transformer layer. This integration preserves the specialization of individual experts while allowing the model to dynamically route tokens across different domains during inference. We denote $\mathrm{FFN}_p^{(l)}(\cdot)$ as the $l$-th FFN sublayer of the $p$-th expert $\mathcal{M}_p$ and let $\mathbf{z}_t^{(l)}$ be the $t$-th token representation of input sequence $\mathbf{x}$ in the $l$-th layer. The corresponding MoE module is then formulated as

$$\mathcal{M}_{\mathrm{MoE}}^{(l)}(\mathbf{z}_t^{(l)}) = \sum_{p \in \mathrm{Top}\text{-}k(\pi^{(l)}(\mathbf{z}_t^{(l)}))} [\pi^{(l)}(\mathbf{z}_t^{(l)})]_p \cdot \mathrm{FFN}_p^{(l)}(\mathbf{z}_t^{(l)}), \tag{5}$$

where $\pi^{(l)}(\mathbf{z}_t)$ is the router's score distribution over experts, and Top-$k(\cdot)$ is the top-$k$ selection.

Conventional routers rely solely on $\mathbf{z}_t^{(l)}$, making routing decisions based on token-level features. However, this can be unreliable: tokens with similar surface forms may belong to different domains and require different experts. Such routing collisions are particularly problematic here, since the router is trained only on proxies and never directly observes the true client data distributions.

To mitigate this, we introduce a context-aware router. Instead of routing purely from $\mathbf{z}_t^{(l)}$, we form a blended representation $\tilde{\mathbf{z}}_t^{(l)} = (1 - \lambda) \, \mathbf{z}_t^{(l)} + \lambda \, \mathbf{z}_{\mathbf{x}}^{(l)}$, where $\mathbf{z}_{\mathbf{x}}^{(l)} = \frac{1}{T} \sum_{t=1}^T \mathbf{z}_t^{(l)}$ is a sequence-level embedding capturing global context ($T$ is the length of $\mathbf{x}$), and $\lambda \in [0, 1]$ is a learnable weight. This blending balances token semantics with broader context cues, and routing distribution is computed as

$$\pi^{(l)}(\mathbf{z}_t^{(l)}) = \mathrm{softmax}[\tilde{\mathbf{z}}_t^{(l)\top} \mathbf{e}_0^{(l)}, \ \tilde{\mathbf{z}}_t^{(l)\top} \mathbf{e}_1^{(l)}, \ \ldots, \ \tilde{\mathbf{z}}_t^{(l)\top} \mathbf{e}_K^{(l)}], \tag{6}$$

where $\mathbf{e}_p^{(l)}$ is the learnable routing vector for expert $p$, initialized as the mean embedding of $\mathcal{D}_p \cup \hat{\mathcal{D}}_p$, i.e., $\mathbf{e}_p^{(l)} = \frac{1}{|\mathcal{D}_p \cup \hat{\mathcal{D}}_p|} \sum_{\mathbf{x} \in \mathcal{D}_p \cup \hat{\mathcal{D}}_p} \mathcal{M}_p^{(1:l)}(\mathbf{x})$, with $\mathcal{M}_p^{(1:l)}(\cdot)$ denoting the first $l$ layers of $\mathcal{M}_p$. This

domain-aware initialization injects each expert's domain characteristics directly into the router, giving it meaningful expert–token priors.

## 3.5 FINAL MOE TRAINING

At the final stage, we aggregate all proxy datasets $\hat{\mathcal{D}}_1 \cup \cdots \cup \hat{\mathcal{D}}_K$ and finetune the unified MoE model. This process updates the router while jointly adapting the FFN experts under supervision from proxy data, ensuring that experts are not treated as isolated components but instead operate cohesively within a MoE architecture. Consequently, the resulting model $\mathcal{M}_{\text{MoE}}$ integrates domain-specific expertise with a privacy-preserving router, yielding a harmonized MoE model that generalizes across heterogeneous client domains. The overall procedure of HarmoMoE is summarized in Algorithm 1.

## 4 EXPERIMENTS

### 4.1 EXPERIMENTS ON CV TASKS

**Datasets.** We evaluate on three benchmarks from distinct domains: (i) Pets [16], with 37 cat and dog breeds, (ii) Flowers [28], with 102 flower categories, and (iii) EuroSAT [10], with 10 land-use classes from satellite imagery. Each dataset serves as a client domain, covering fine-grained categorization, natural object recognition, and remote sensing. As the public dataset $\mathcal{D}_0$, we adopt ImageNet [3], which is a large-scale visual corpus providing 1.28M images across 1,000 categories.

**Models.** We adopt two vision–language models CLIP ViT-B/32 and ViT-B/16 [31] as the seed models, which consist of a transformer-based image encoder and a text encoder.

**Implementation Details.** For each client, we build a proxy dataset by first selecting a candidate pool of $n = 3000$ ImageNet samples that are most similar to its private data according to the relevance score $g(\mathbf{x}, \mathcal{D}_p)$, then choosing $m = 500$ proxy samples via relevance-weighted DPP with cosine similarity kernel $\kappa(\mathbf{x}_i, \mathbf{x}_j) = \cos(\mathbf{z}_i, \mathbf{z}_j)$, where $\mathbf{z}_i$ is $\mathbf{x}_i$'s embedding extracted from $\mathcal{M}_0$. Client models are initialized from the seed model $\mathcal{M}_0$ and finetuned on both private and proxy data. For proxy-aligned expert training, we finetune the FFN sublayers of the visual encoder using LoRA [12] (rank 16, scaling factor $\alpha = 32$) for 10 epochs with the SGD optimizer (momentum 0.9, weight decay 0.0001, learning rate 0.01, batch size 128, constant learning rates schedule). For router training, we adopt top-1 routing and finetune for 5 epochs using SGD optimizer with a learning rate of 0.001.

**Baselines.** We compare our method against various baselines: (i) ZeroShot directly evaluates the seed model $\mathcal{M}_0$ without any adaptation, providing a capacity-only lower bound. (ii) BTM [20] ensembles predictions from independently trained experts at inference, but does not produce a unified model for downstream use. (iii) ModelSoup [45] merges experts by averaging their weights into a single model, avoiding inference overhead from ensembling but losing specialization when experts diverge. (iv) BTX (Branch-Train-MiX) [39] integrates experts by inserting their FFN sublayers into MoE layers, averaging the remaining parameters, and then fine-tuning the mixed model on public data (since private data are unavailable), following the setup of FlexOlmo [38]. (v) FlexOlmo [38] aligns experts without centralizing private data by anchoring them to a shared public model and introducing per-expert router embeddings, which are later finetuned on proxy data selected by similarity to private domains. (vi) Separate Experts evaluates each independently finetuned expert across all domains. (vii) UnrestrictedMoE trains a unified MoE model directly on the merged private datasets from all clients, thereby achieving strong performance but breaking privacy constraints.

**Results.** Tables 1 and 2 report the testing accuracy on the three client domains when using CLIP ViT-B/32 and CLIP ViT-B/16 as the seed model, respectively. Across both backbones, our proposed HarmoMoE consistently outperforms all privacy-preserving baselines. Compared with BTM, which ensembles predictions from independent experts without parameter sharing, HarmoMoE integrates experts at the parameter level and leverages a router to coordinate experts, thereby achieving better performance. Compared with ModelSoup—which averages model parameters—HarmoMoE achieves much higher accuracy by explicitly retaining expert specialization within a unified MoE architecture. In contrast to BTX and FlexOlmo, which also employ MoE-style integration but rely on proxy data without explicit distribution alignment, HarmoMoE yields clear gains by combining relevance-weighted DPP-selected proxy subsets with proxy-aligned expert training, allowing routers to coordinate experts more effectively. Quantitatively, HarmoMoE achieves an average accuracy of

Table 1: Accuracy of CV Tasks when using CLIP ViT-B/32 as the seed model.

|  | Pets | Flowers | EuroSAT | **Average** |
|---|---|---|---|---|
| UnrestrictedMoE | 92.45 | 96.43 | 98.15 | 95.68 |
| ZeroShot | 85.77 | 61.59 | 29.81 | 59.06 |
| Expert I (Pets) | 92.40 | 59.03 | 22.74 | 58.06 |
| Expert II (Flowers) | 84.25 | 96.91 | 27.21 | 69.46 |
| Expert III (EuroSAT) | 82.64 | 52.42 | 97.91 | 77.66 |
| BTM [20] | 90.81 | 85.10 | 95.07 | 90.33 |
| ModelSoup [45] | 87.90 | 70.52 | 64.19 | 74.20 |
| BTX [39] | 88.44 | 75.07 | 59.38 | 74.30 |
| FlexOlmo [38] | 91.36 | 90.62 | 96.79 | 92.92 |
| HarmoMoE | **91.91** | **93.67** | **97.98** | **94.52** |

Table 2: Testing Accuracy of CV Tasks when using CLIP ViT-B/16 as the seed model.

|  | Pets | Flowers | EuroSAT | **Average** |
|---|---|---|---|---|
| UnrestrictedMoE | 94.30 | 97.73 | 98.23 | 96.75 |
| ZeroShot | 88.53 | 68.09 | 34.00 | 63.54 |
| Expert I (Pets) | 94.44 | 65.69 | 30.10 | 63.41 |
| Expert II (Flowers) | 85.99 | 98.13 | 24.05 | 69.39 |
| Expert III (EuroSAT) | 86.56 | 60.05 | 98.43 | 81.68 |
| BTM [20] | 93.21 | 84.65 | 97.38 | 91.75 |
| ModelSoup [45] | 89.94 | 74.18 | 74.14 | 79.42 |
| BTX [39] | 89.89 | 78.52 | 75.20 | 81.20 |
| FlexOlmo [38] | 94.09 | 89.40 | 97.11 | 93.53 |
| HarmoMoE | **94.22** | **97.08** | **97.41** | **96.24** |

94.52% with CLIP ViT-B/32 and 96.24% with CLIP ViT-B/16, outperforming the strongest baseline FlexOlmo (92.92% and 93.53%) by about 1.6 and 2.7 points, respectively.

## 4.2 EXPERIMENTS ON NLP TASKS

**Datasets.** We evaluate on three benchmarks for commonsense reasoning: (i) CommonsenseQA [40], which tests general world knowledge through discrimination among semantically related concepts, (ii) CosmosQA [13], which emphasizes narrative reasoning by requiring inference of implicit causes and effects, and (iii) SocialIQA [35], which focuses on social commonsense involving human actions, motivations, and social implications. Proxy data are selected from Alpaca [41], a publicly available collection of 52K instruction–response pairs that spans diverse instruction-tuning tasks. Note that the public dataset (Alpaca) has no domain overlap with the client datasets.

**Models.** We evaluate on two LLaMA models across different scales: LLaMA-3.2-3B [27], a lightweight 3B model for efficiency-critical settings, and LLaMA-3.1-8B [26], a larger 8B model offering stronger performance with higher compute cost.

**Implementation Details.** For each client, we form a proxy dataset by first selecting a candidate pool of $n = 3000$ Alpaca samples most similar to its private data, then choosing $m = 500$ proxies via relevance-weighted DPP with cosine similarity kernel $\kappa(\mathbf{x}_i, \mathbf{x}_j) = \cos(\mathbf{z}_i, \mathbf{z}_j)$, where $\mathbf{z}_i$ is $\mathbf{x}$'s embedding extracted from $\mathcal{M}_0$. Client models are initialized from the seed model $\mathcal{M}_0$. For proxy-aligned expert training, we finetune the model's FFN sublayers on both private and proxy data using LoRA [12] (rank 16, $\alpha = 32$) for 10 epochs with AdamW [23] (learning rate 0.0001, batch size 32, no weight decay, constant schedule with 200 warm-up steps). For router training, we adopt top-1 routing and finetune for 1 epoch with AdamW (learning rate 0.0001, batch size 32, no weight decay).

**Results.** Tables 3 and 4 report the testing accuracy of NLP tasks with LLaMA-3.2-3B and LLaMA-3.1-8B as the seed models, respectively. Across both backbones, HarmoMoE consistently outperforms all privacy-preserving baselines. Unlike BTM, which ensembles outputs without parameter sharing, or ModelSoup, which only averages parameters, HarmoMoE preserves expert specialization within

Table 3: Accuracy of NLP Tasks when using LLaMA-3.2-3B as the seed model.

| | CommonsenseQA | CosmosQA | SocialIQA | **Average** |
|---|---|---|---|---|
| UnrestrictedMoE | 75.51 | 78.39 | 71.80 | 75.23 |
| ZeroShot | 62.49 | 62.68 | 56.19 | 60.45 |
| Expert I (CommonsenseQA) | 74.94 | 70.18 | 60.54 | 68.55 |
| Expert II (CosmosQA) | 63.06 | 78.69 | 58.96 | 66.90 |
| Expert III (SocialIQA) | 65.44 | 68.04 | 72.42 | 68.63 |
| BTM [20] | 74.61 | 75.44 | 68.17 | 72.74 |
| ModelSoup [45] | 73.71 | 75.24 | 71.75 | 73.57 |
| BTX [39] | 69.62 | 72.06 | 71.75 | 71.14 |
| FlexOlmo [38] | 73.30 | 73.33 | 70.88 | 72.50 |
| HarmoMoE | **74.94** | **76.05** | **72.26** | **74.42** |

Table 4: Accuracy of NLP Tasks when using LLaMA-3.1-8B as the seed model.

| | CommonsenseQA | CosmosQA | SocialIQA | **Average** |
|---|---|---|---|---|
| UnrestrictedMoE | 81.90 | 85.70 | 76.41 | 81.34 |
| ZeroShot | 69.37 | 75.98 | 57.57 | 67.64 |
| Expert I (CommonsenseQA) | 81.24 | 77.29 | 69.19 | 75.91 |
| Expert II (CosmosQA) | 71.42 | 86.23 | 67.86 | 75.17 |
| Expert III (SocialIQA) | 73.63 | 76.98 | 78.10 | 76.24 |
| BTM [20] | 81.24 | 80.84 | 77.28 | 79.79 |
| ModelSoup [45] | 80.26 | 84.25 | 77.02 | 80.51 |
| BTX [39] | 75.76 | 81.04 | 73.39 | 76.73 |
| FlexOlmo [38] | 75.18 | 81.11 | 76.10 | 77.46 |
| HarmoMoE | **81.33** | **85.80** | **77.64** | **81.59** |

a unified MoE model and enables router-based coordination. Compared with BTX and FlexOlmo, which also use proxy data but lack relevance-diverse alignment mechanisms, HarmoMoE further benefits from DPP-selected proxies that approximate private distributions more accurately, leading to more effective routing. Quantitatively, HarmoMoE achieves $74.42\%$ average accuracy on LLaMA-3.2-3B and $81.59\%$ on LLaMA-3.1-8B, outperforming the strongest baselines at both scales.

## 4.3 ABLATION STUDY: USEFULNESS OF RELEVANCE-WEIGHTED DPP

We conduct an ablation study to evaluate the effect of incorporating relevance-weighted DPP into proxy data selection (Section 3.2) for HarmoMoE and the recent method FlexOlmo. The configuration denoted as "✗" in Table 5 corresponds to FlexOlmo's original similarity-based approach, which selects public samples based on their relevance scores to the client's private data, without enforcing diversity. As shown in Table 5, DPP consistently boosts accuracy across both NLP and CV tasks, highlighting its effectiveness as a selection strategy. For FlexOlmo, incorporating DPP leads to clear accuracy gains across all tasks (e.g., +0.85 on LLaMA-3.2-3B and +2.32 on LLaMA-3.1-8B), demonstrating that existing methods can benefit significantly from our proposed relevance-weighted DPP proxy selection. HarmoMoE further boosts these benefits, with consistent improvements in all settings, ultimately achieving the best overall performance. These findings show that relevance-weighted DPP is crucial for selecting proxy samples that are effective for training the router to coordinate experts.

Table 5: Accuracy of FlexOlmo and HarmoMoE with and without relevance-weighted DPP.

| | Relevance-Weighted DPP | NLP | | CV | |
|---|---|---|---|---|---|
| | | LLaMA-3.2-3B | LLaMA-3.1-8B | ViT-B/32 | ViT-B/16 |
| FlexOlmo | ✗ | 72.50 | 77.46 | 92.92 | 93.53 |
| | ✓ | 73.35 | 79.78 | 93.20 | 94.38 |
| HarmoMoE | ✗ | 73.60 | 80.32 | 94.12 | 95.39 |
| | ✓ | **74.42** | **81.59** | **94.52** | **96.24** |

**Visualization of Selected Proxy Samples.** In Figure 2, we use t-SNE [42] to visualize the proxy samples selected by three different selection strategies: *random sampling*, *similarity-based selection* used in FlexOlmo, and our HarmoMoE based on *relevance-weighted DPP*. As can be seen from Figure 2(a), random selection fails to capture either relevance or diversity, often yielding samples that are poorly aligned with the private data distribution. FlexOlmo improves alignment by selecting samples highly similar to the private domain, but the resulting proxies lack diversity: many chosen samples cluster in a narrow region of the data space, providing limited coverage of the right side of the private-data manifold (Figure 2(b)). In contrast, our HarmoMoE explicitly balances relevance and diversity through the relevance-weighted DPP kernel. As shown in Figure 2(c), the resulting proxy set not only anchors closely to the private domain but also forms a diverse cover of the data space. This richer proxy distribution allows the router to discriminate across heterogeneous contexts. Consequently, our HarmoMoE provides a stronger and more aligned proxy of the unavailable private data, which translates into consistent performance gains (Table 5).

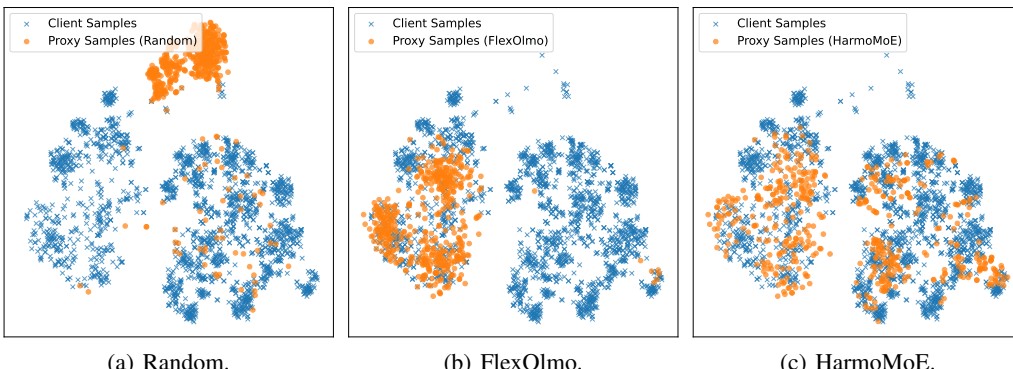

|        (a) Random.        |        (b) FlexOlmo.        |        (c) HarmoMoE.        |

Figure 2: t-SNE visualization of selected proxy samples with random selection, FlexOlmo selection (relevance only), and our selection (relevance + diversity) for Pets with ViT-B/32 as the seed model.

## 4.4 ABLATION STUDY: USEFULNESS OF THE CONTEXT-AWARE ROUTER

We examine the effectiveness of the proposed context-aware router (Section 3.4) across both CV and NLP tasks. As shown in Table 7, incorporating sequence-level context consistently yields higher accuracy across all evaluated datasets and backbones. Specifically, average accuracy improves by roughly 0.5–1.0 points on vision benchmarks (e.g., from 93.9 to 94.5 on ViT-B/32 and from 95.9 to 96.2 on ViT-B/16) and by about 1.5–2.0 points on language benchmarks (e.g., from 72.6 to 74.4 on LLaMA-3.2-3B and from 79.4 to 81.6 on LLaMA-3.1-8B). These consistent gains demonstrate that incorporating sequence-level context into the router enables more reliable expert assignment.

Table 6: Accuracy of HarmoMoE with and without context-aware router for CV tasks.

|           | Context-Aware Router | Pets | Flowers | EuroSAT | **Average** |
|-----------|:---:|:---:|:---:|:---:|:---:|
| ViT-B/32  | ✗ | 91.69 | 93.18 | 96.90 | 93.92 |
|           | ✓ | **91.91** | **93.67** | **97.98** | **94.52** |
| ViT-B/16  | ✗ | 93.95 | 96.71 | 97.16 | 95.94 |
|           | ✓ | **94.22** | **97.08** | **97.41** | **96.24** |

Table 7: Accuracy of HarmoMoE with and without context-aware router for NLP tasks.

|              | Context-Aware Router | CommonsenseQA | CosmosQA | SocialIQA | **Average** |
|--------------|:---:|:---:|:---:|:---:|:---:|
| LLaMA-3.2-3B | ✗ | 73.63 | 72.16 | 72.06 | 72.62 |
|              | ✓ | **74.94** | **76.05** | **72.26** | **74.42** |
| LLaMA-3.1-8B | ✗ | 80.51 | 80.34 | 77.38 | 79.41 |
|              | ✓ | **81.33** | **85.80** | **77.64** | **81.59** |

## 4.5 ABLATION STUDY: USEFULNESS OF PROXY-ALIGNED EXPERT TRAINING

We assess the role of proxy-aligned expert training (Section 3.3) by comparing experts trained only on private data with those additionally exposed to client-specific proxy data. Experiments are conducted on both CV and NLP tasks. As shown in Table 8, incorporating proxy data yields consistent gains across both ViT-B/32 and ViT-B/16 on CV tasks. Similarly, on NLP tasks (see Table 9), proxy-aligned training improves average accuracy from 72.71 to 74.42 on LLaMA-3.2-3B and from 80.99 to 81.59 on LLaMA-3.1-8B. These results suggest that without proxy data, experts may become misaligned with the router, while proxy-aligned training calibrates them to the same supervision as the router, enabling more effective expert collaboration.

Table 8: Performance of HarmoMoE with and without Proxy-Aligned Expert Training for CV tasks.

|  | Proxy-Aligned Training | Pets | Flowers | EuroSAT | **Average** |
|---|---|---|---|---|---|
| ViT-B/32 | ✗ | 91.44 | 91.92 | 95.57 | 92.98 |
|  | ✓ | **91.91** | **93.67** | **97.98** | **94.52** |
| ViT-B/16 | ✗ | 94.22 | 90.70 | 97.35 | 94.09 |
|  | ✓ | **94.22** | **97.08** | **97.41** | **96.24** |

Table 9: Performance of HarmoMoE with and without Proxy-Aligned Expert Training for NLP tasks.

|  | Proxy-Aligned Training | CommonsenseQA | CosmosQA | SocialIQA | **Average** |
|---|---|---|---|---|---|
| LLaMA-3.2-3B | ✗ | 74.77 | 71.56 | 71.80 | 72.71 |
|  | ✓ | **74.94** | **76.05** | **72.26** | **74.42** |
| LLaMA-3.1-8B | ✗ | 80.75 | 84.89 | 77.33 | 80.99 |
|  | ✓ | **81.33** | **85.80** | **77.64** | **81.59** |

## 5 CONCLUSION

We proposed HarmoMoE, a harmonized mixture-of-experts framework for integrating independently trained experts under privacy constraints. By leveraging a relevance-weighted DPP for proxy data selection, proxy-aligned expert training, and a context-aware router, our method enables effective routing without direct access to private data. Experiments on CV and NLP benchmarks show that HarmoMoE consistently outperforms recent state-of-the-art methods, suggesting a promising path toward scalable and privacy-preserving MoE systems in collaborative and decentralized environments.

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

## A    Computation of Relevance Score

Following FlexOlmo [38], we compute the relevance score $g(\mathbf{x}, \mathcal{D}_p)$ of a public sample $\mathbf{x} \in \mathcal{D}_0$ with respect to a client dataset $\mathcal{D}_p$ by training a binary classifier to distinguish $\mathcal{D}_p$ from $\mathcal{D}_0$. Specifically, we construct a training set by labeling samples from $\mathcal{D}_p$ as positive and randomly drawing 10K samples from $\mathcal{D}_0$ as negative. Specifically, we append a classification head to the last hidden layer of the seed model, and finetune this classifier to distinguish whether a sample comes from the client dataset $\mathcal{D}$ (positive) or the public dataset $\mathcal{D}_0$ (negative). After training, we apply the classifier to every $\mathbf{x} \in \mathcal{D}_0$. The predicted probability that $\mathbf{x}$ belongs to $\mathcal{D}_p$ is used as the relevance score:

$$g(\mathbf{x}, \mathcal{D}_p) = \mathbb{P}[\text{classifier predicts } \mathbf{x} \in \mathcal{D}_p]. \tag{7}$$

This score quantifies the extent to which each public sample is representative of the private domain. In practice, we rank all $\mathbf{x} \in \mathcal{D}_0$ by $g(\mathbf{x}, \mathcal{D}_p)$ and retain the top-$n$ candidates for subsequent proxy selection.

## B    Cholesky Updates for Efficient Inference

In this appendix, we show how to make greedy DPP MAP inference computationally efficient for large-scale proxy selection. The main challenge is the repeated evaluation of determinants for growing kernel submatrices. A naive implementation recomputes a full factorization for each of the $n$ candidates at every greedy step, incurring $O(nm^3)$ time per iteration when the current subset has size $m$. We address this bottleneck by maintaining a Cholesky factorization of the current kernel submatrix and updating it incrementally as new elements are considered. With this strategy, scoring all $n$ candidates in a greedy iteration requires only $O(nm)$ time. The derivation below details this update mechanism and explains its implications for scalability.

**Naive Computation.**    Consider a candidate subset $\mathcal{S} \subseteq \{1, \dots, n\}$ with associated kernel submatrix $\widetilde{\mathbf{L}}_\mathcal{S}$. At each step of greedy MAP inference, one must compute

$$\det(\widetilde{\mathbf{L}}_{\mathcal{S} \cup \{\mathbf{x}\}})$$

for a new candidate $\mathbf{x} \notin \mathcal{S}$. If performed directly, this requires recomputing the determinant of an $(|\mathcal{S}| + 1) \times (|\mathcal{S}| + 1)$ matrix, incurring $O(m^3)$ time per evaluation. Over all $n$ candidates and $m$ greedy steps, the total cost for each greedy step scales as $O(nm^3)$, which is infeasible when both $n$ and $m$ are large.

**Cholesky Factorization.**    To avoid redundant recomputation, we exploit the Cholesky decomposition [11]. Suppose the current kernel submatrix admits a decomposition

$$\widetilde{\mathbf{L}}_\mathcal{S} = \mathbf{P}\mathbf{P}^\top,$$

where $\mathbf{P}$ is a lower-triangular matrix of size $|\mathcal{S}| \times |\mathcal{S}|$. The determinant is then easily obtained as

$$\det(\widetilde{\mathbf{L}}_\mathcal{S}) = \prod_{i=1}^{|\mathcal{S}|} \mathbf{P}_{ii}^2,$$

so that the computational burden shifts from determinant computation to maintaining $\mathbf{P}$.

**Incremental Update.**    When a new element $\mathbf{x}$ is considered, the augmented kernel matrix can be written in block form:

$$\widetilde{\mathbf{L}}_{\mathcal{S} \cup \{\mathbf{x}\}} = \begin{bmatrix} \widetilde{\mathbf{L}}_\mathcal{S} & \mathbf{k} \\ \mathbf{k}^\top & \widetilde{\mathbf{L}}_{\mathbf{xx}} \end{bmatrix}, \tag{8}$$

where $\mathbf{k}$ contains the similarities between $\mathbf{x}$ and items in $\mathcal{S}$. Instead of recomputing a full factorization, we extend $\mathbf{P}$ by one row and column:

$$\mathbf{P}' = \begin{bmatrix} \mathbf{P} & 0 \\ \mathbf{y}^\top & \sigma \end{bmatrix},$$

so that $\mathbf{P}'\mathbf{P}'^{\top} = \widetilde{\mathbf{L}}_{\mathcal{S}\cup\{\mathbf{x}\}}$. Expanding both sides and equating with (8) gives

$$\mathbf{P}\mathbf{y} = \mathbf{k}, \qquad \sigma^2 = \widetilde{\mathbf{L}}_{\mathbf{xx}} - \|\mathbf{y}\|_2^2.$$

Thus, the update reduces to solving a triangular system (for $\mathbf{y}$) and computing a residual variance (for $\sigma^2$). Both steps are efficient: solving a triangular system costs $O(m)$, and computing a norm is linear in $m$ as well. Hence, each iteration of greedy MAP inference only requires a total time of $\mathcal{O}(nm)$ for searching over the $n$ candidates. This makes the approach scalable to large public datasets, while still retaining the DPP's balance between relevance and diversity.

## C  COMPUTER VISION DATASETS

Figure 3 presents randomly sampled images from the three client datasets used in the CV experiments: Pets [16], Flowers [28], and EuroSAT [10]. These examples illustrate the visual diversity across domains, ranging from fine-grained object recognition of dog and cat breeds (Pets), to natural scene categorization of flower species (Flowers), and remote sensing imagery for land-use classification (EuroSAT). Such heterogeneity highlights the challenge of unifying domain-specialized experts into a single model while preserving privacy and ensuring robust multi-domain generalization.

We adopt ImageNet [3] as the public dataset from which proxy samples are drawn. Figure 4 shows randomly sampled examples from ImageNet.

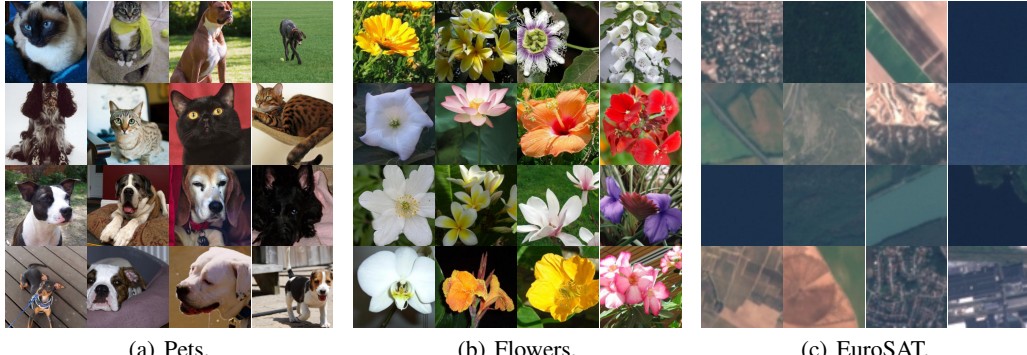

(a) Pets.                    (b) Flowers.                    (c) EuroSAT.

Figure 3: Sample images from the three client domains: Pets, Flowers, and EuroSAT.

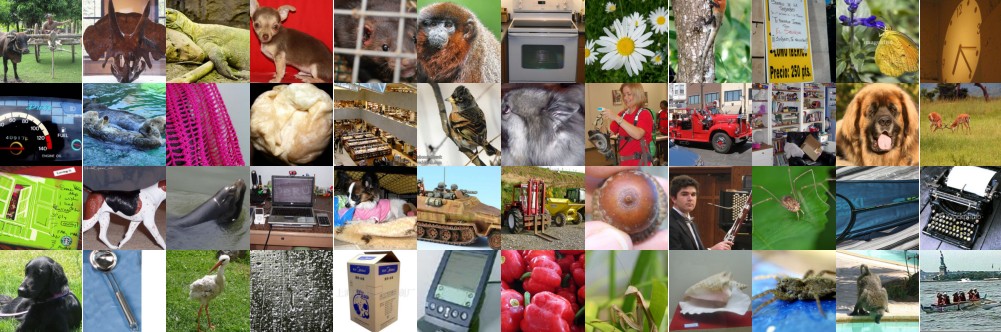

Figure 4: Sample images from ImageNet.

## D  NATURAL LANGUAGE PROCESSING DATASETS

The client-side NLP datasets comprise CommonsenseQA [40], CosmosQA [13], and SocialIQA [35]. These cover complementary reasoning skills: CommonsenseQA requires grounding abstract questions in everyday knowledge; CosmosQA emphasizes multi-sentence comprehension with causal and temporal reasoning; and SocialIQA targets social motivations and reactions in human interactions. Examples 1–3 show two representative samples from CommonsenseQA, CosmosQA, and SocialIQA,

respectively. Such diversity highlights the challenge of integrating domain-specialized experts in NLP while ensuring broad generalization across reasoning styles and task formats.

As the public corpus $\mathcal{D}_0$, we use Alpaca [41], an open-domain instruction–response dataset ranging from factual queries to reasoning and generation. Example 4 illustrates three instances from Alpaca.

---

**Example 1: Samples from CommonsenseQA [40]**

**Question:** The fox walked from the city into the forest, what was it looking for?
(A) pretty flowers
(B) hen house
(C) natural habitat
(D) storybook
(E) dense forest
**Answer:** (C) natural habitat

**Question:** To learn one must have the right book, to work efficiently what must one have?
(A) improve yourself
(B) become knowledgeable
(C) have tools
(D) persistence
(E) have more knowledge
**Answer:** (C) have tools

---

**Example 2: Samples from CosmosQA [13]**

Let me be clear, everything is good. Having said that, I 've been a little preoccupied lately, as Leslie threw out her back on Sunday. It's been an interesting, and marriage-solidifying 36 hours. Sunday afternoon, around 2 pm, Leslie and I were hanging in the library, sans kids (they were with their father), listening to music and talking.
**Question:** How did Leslie's throwing out of her back help to solidify your marriage?
(A) Leslie realized that I am really good at taking care of people and wants to keep me around for that reason
(B) Leslie saw that if she ever hurt herself she would need my help to heal and realized that she needs me
(C) None of the above choices
(D) We were able to spend time together just talking and that helped us to connect better
**Answer:** (D) We were able to spend time together just talking and that helped us to connect better

But the thing is, I forgave and forgot a while ago. I 'm not bitter anymore. I just genuinely don't want to go back to that church. I couldn't get the balls to tell her that ...
**Question:** Why wouldn't I have the balls to tell her that?
(A) I am not a man and I can't tell her those things to her face
(B) I am afraid of that church and I would feel less manly if I told her
(C) None of the above choices
(D) It is her favorite church and I don't want to hurt her feelings
**Answer:** (D) It is her favorite church and I don't want to hurt her feelings

---

**Example 3: Samples from SocialIQA [35]**

During their yearbook class, Alex took an old photo of herself and Quinn, and crossed out Quinn's face on the photograph. Quinn saw her do this.
**Question:** How would Quinn feel as a result?
(A) Like Alex misses hanging out with her
(B) Like Alex betrayed her
(C) be friends again
**Answer:** (B) Like Alex betrayed her

Tracy was teaching Kendall to dance so Tracy raised Kendall's hips into the right position for the next move.
**Question:** How would you describe Tracy?
(A) a bad teacher
(B) hopeful that this doesn't make Kendall uncomfortable
(C) a patient teacher
**Answer:** (B) hopeful that this doesn't make Kendall uncomfortable

---

> **Example 4: Samples from Alpaca [41]**
>
> **Question:** Use the given data to calculate the median.
> **Input:** [2, 3, 7, 8, 10]
> **Answer:** The median is 7.
>
> **Question:** Analyze the given text for its tone.
> **Input:** The world has been greatly impacted by the COVID-19 pandemic and it has drastically changed our lives.
> **Answer:** The tone of the text is one of concern and recognition of the dramatic effects of the COVID-19 pandemic on the world. It is an acknowledgement of the upheaval that the virus has caused and that our lives have been drastically changed.
>
> **Question:** Explain the concept of artificial intelligence in simple terms.
> **Answer:** Artificial Intelligence (AI) is the ability of machines and computers to make decisions and complete tasks similar to what humans can do. AI can learn from data, recognize patterns, and make predictions. AI technology can automate tedious and complex processes, helping to reduce human effort and errors, and enable more accurate and efficient decisions. AI can be used in a wide range of applications, from robotics and autonomous vehicles to healthcare. It is increasingly becoming an integral part of everyday life.

## E  LARGE LANGUAGE MODEL USAGE STATEMENT

In preparing this manuscript, we used large language models (LLMs) solely as tools for polishing the writing and improving clarity of presentation. All ideas, methodological designs, theoretical derivations, and experimental results are original to the authors. LLMs were not involved in generating research content or making scientific contributions, and they are not considered authors of this work.

## F  ETHICS STATEMENT

This paper addresses the challenge of unifying domain-specialized experts into a single Mixture-of-Experts (MoE) model under strict privacy constraints. We emphasize that our framework does not require sharing client data and therefore avoids risks of exposing sensitive or proprietary information. All experiments are conducted using publicly available datasets, and no personally identifiable information or human subjects are involved.

Our methodology does not generate or promote harmful content, and it is intended for responsible applications in settings where privacy and collaboration are both essential. We have no conflicts of interest, and our research complies with the ICLR Code of Ethics. We encourage the community to apply this framework responsibly, particularly in sensitive or high-stakes domains.

## G  REPRODUCIBILITY STATEMENT

We have taken multiple steps to ensure reproducibility of our work. The proposed HarmoMoE framework is described in detail in Section 3, with all algorithmic steps summarized in Algorithm 1. Datasets and preprocessing procedures are specified in Section 4. Experimental settings, including hyperparameters, model backbones, and training details, are reported in Section 4. After the paper is accepted, we will release source code, including scripts for proxy selection, expert training, and router integration in the final version.

## H  ADDITIONAL EXPERIMENTS

### H.1  COMPUTATION OVERHEAD OF RELEVANCE-WEIGHTED DPP

Both FlexOlmo and HarmoMoE require computing embeddings for all public samples, which dominates the proxy-selection cost. The additional work unique to HarmoMoE is the greedy MAP inference for relevance-weighted DPP, implemented with efficient Cholesky updates (see Appendix B). Table 10 shows the measured running time for NLP and CV setups. Relevance-weighted DPP adds only 0.2–0.3 minutes (3–5%) per client, while expert finetuning takes over 10 GPU-hours, confirming

that the diversity-aware selection introduces only marginal overhead relative to similarity-only baselines while delivering more representative proxy sets.

Table 10: Proxy-selection running time (minutes).

|  | NLP | | CV | |
| --- | --- | --- | --- | --- |
|  | LLaMA-3.2-3B | LLaMA-3.1-8B | ViT-B/32 | ViT-B/16 |
| Similarity-based selection | 1.91 | 4.21 | 6.41 | 15.70 |
| Relevance-weighted DPP | 2.17 | 4.54 | 6.61 | 15.91 |

## H.2 COMPARISON WITH CoMiGS [5] AND MoA [7]

Federated learning (FL) methods differ fundamentally from HarmoMoE: they require synchronous parameter aggregation, exchanging large model states every round. This leads to substantial bandwidth and memory overhead and creates instability when client data are heterogeneous because divergent local updates must be averaged. HarmoMoE eliminates synchronization entirely—each client fine-tunes its expert locally (on private plus proxy data), and only a single exchange of frozen expert weights occurs before router training.

**Comparison with CoMiGS [5].** CoMiGS [5] adopts a federated learning paradigm that requires repeated synchronization among clients—periodically exchanging model parameters for joint optimization. This approach incurs high communication and memory costs and often becomes unstable under heterogeneous client data, leading to degraded performance. In contrast, HarmoMoE removes synchronization entirely. Each client independently fine-tunes its expert on private and proxy data, and all experts are unified once through MoE integration. This asynchronous, merge-after-training design achieves better scalability, eliminates communication overhead, and remains stable under heterogeneous data.

**Comparison with Mixture-of-LoRAs (MoA) [7].** MoA also seeks to unify multiple LoRA experts but assumes direct access to all client data for router training, which violates privacy constraints. In the experiments reported here, MoA is reimplemented by training its router only on public data to maintain expert coordination without violating privacy constraints, enabling a fair comparison under the same privacy-preserving setting.

**Empirical results.** HarmoMoE is compared with CoMiGS and MoA under the same LoRA configurations. Due to memory constraints, CoMiGS could not scale to LLaMA-3.1-8B (out-of-memory issues). As shown in Tables 11 and 12, HarmoMoE consistently outperforms both methods across all benchmarks, demonstrating that HarmoMoE's unification offers better performance.

Table 11: Comparison with CoMiGS and MoA on NLP tasks (LLaMA-3.2-3B).

|  | CommonsenseQA | CosmosQA | SocialIQA | Average |
| --- | --- | --- | --- | --- |
| CoMiGS [5] | 72.32 | 71.46 | 71.19 | 71.65 |
| MoA [7] | 71.09 | 74.64 | 70.11 | 71.95 |
| HarmoMoE | **74.94** | **76.05** | **72.26** | **74.42** |

Table 12: Comparison with CoMiGS and MoA on NLP tasks (LLaMA-3.1-8B).

|  | CommonsenseQA | CosmosQA | SocialIQA | Average |
| --- | --- | --- | --- | --- |
| CoMiGS [5] | — | — | — | — |
| MoA [7] | 77.72 | 81.64 | 74.05 | 77.81 |
| HarmoMoE | **81.33** | **85.80** | **77.64** | **81.59** |

### H.3    TRAINING SOLELY ON PROXY DATA

To isolate the role of private data, we conduct an ablation where experts are trained solely on proxy samples and never observe client data. We evaluate this proxy-only baseline in the CLIP ViT-B/32 CV setting. As summarized in Table 13, training exclusively on proxies leads to large accuracy drops, confirming that proxies primarily provide coordination signals for router learning. HarmoMoE, which blends private data (for domain expertise) with proxies (for alignment), substantially outperforms the proxy-only alternative, demonstrating the necessity of private-domain expertise.

Table 13: Proxy-only baseline versus HarmoMoE (CLIP ViT-B/32).

|  | Pets | Flowers | EuroSAT | Average |
|---|---|---|---|---|
| Train solely on proxy data | 78.60 | 42.63 | 12.98 | 44.74 |
| HarmoMoE | **91.91** | **93.67** | **97.98** | **94.52** |

### H.4    EFFECT OF FINAL FINE-TUNING

We ablate the final fine-tuning stage, where the unified MoE (experts and router) is jointly optimized on the aggregated proxy data. Without this step, experts retain their individual adaptations and the router remains partially misaligned, degrading coordination. Table 14 reports results on CLIP ViT-B/32, showing that final fine-tuning yields an average improvement of +11 percentage points, underscoring its importance for coordinated expert collaboration in router training.

Table 14: Impact of final fine-tuning (CLIP ViT-B/32).

|  | Pets | Flowers | EuroSAT | Average |
|---|---|---|---|---|
| w/o final fine-tuning | 86.78 | 80.35 | 81.86 | 83.00 |
| w/ final fine-tuning | **91.91** | **93.67** | **97.98** | **94.52** |

### H.5    PRIVACY-PRESERVING GUARANTEES

HarmoMoE never exposes private data, and any similarity between selected proxy samples and private data does not constitute a privacy violation, as all proxy candidates are drawn from a public dataset that is already accessible to all parties.

**(1) Semantic similarity is not equivalent to private-data exposure.** The relevance-weighted DPP method selects public samples that are representation-wise similar to the client domain. These proxies may resemble private data but are not derived from private samples, and all clients already have access to them. In privacy-preserving systems, leakage occurs only when *non-public information becomes newly revealed*. Selecting an already public sample—no matter how similar—does not expose any new private information.

**(2) Overlap with public data does not constitute private-data exposure.** If a public sample coincidentally overlaps with one in a client's dataset, revealing that sample still constitutes *public-data exposure*, not private-data exposure, since the content is already publicly available prior to any interaction with HarmoMoE. This principle aligns with standard privacy frameworks such as the California Consumer Privacy Act (CCPA), which explicitly excludes "publicly available information" from the definition of personal data. Under this definition, *the exposure of a public sample is not regarded as a violation of privacy*.

**(3) HarmoMoE introduces no new channels for private-data leakage.** The proxy-selection process transmits only the IDs of selected public samples and the final expert weights—never private samples, gradients, or intermediate activations. Because all proxy candidates are drawn from a public dataset, the process does not disclose or allow inference about private data.

## H.6 COMPARISON WITH LORASUITE [22]

HarmoMoE is a low-rank adaptation unification method, whereas LoRASuite [22] is not, and the two address fundamentally different goals. HarmoMoE aims to *unify multiple domain-specialized LoRA experts* trained on the same backbone into a privacy-preserving MoE—answering how to combine many LoRA experts without sharing client data. In contrast, LoRASuite focuses on *LoRA migration*, transferring a single LoRA adapter trained on backbone (A) to backbone (B) after the backbone is upgraded, with the goal of expert adaptation rather than expert unification. Furthermore, LoRASuite requires *client data* to align activations and performs poorly without it, while HarmoMoE assumes *no private data access* and relies entirely on public proxies for supervision. Given these fundamental differences in objectives and data requirements, a direct empirical comparison would be inappropriate, as it would force LoRASuite into a multi-expert unification setting it was never designed for.

## H.7 COMPUTATIONAL COST ANALYSIS

This section provides a detailed comparison of HarmoMoE's computational cost against existing methods, covering both unification time and inference speed across vision and language tasks. The results demonstrate that HarmoMoE achieves strong accuracy improvements without adding significant overhead.

**Unification cost.** The merging (unification) time of HarmoMoE is nearly identical to FlexOlmo and BTX across all backbones, showing that the accuracy gains do not come from higher computational cost during unification. While BTM and ModelSoup appear faster (or cost-free), they avoid the coordination required for MoE merging, which explains their lower accuracy. The small extra cost in HarmoMoE is therefore a modest and worthwhile trade-off for its significant accuracy improvement.

**Inference efficiency.** HarmoMoE maintains inference speeds comparable to other MoE unification methods, confirming that the context-aware router introduces minimal runtime overhead and scales efficiently across backbones. In contrast, BTM requires inference over all experts for every input, leading to approximately $3\times$ slower inference despite its zero unification cost.

**Overall cost–performance balance.** Across both CV and NLP tasks, HarmoMoE consistently achieves state-of-the-art accuracy with comparable computational efficiency, as summarized in Tables 15 and 16.

Table 15: Cost–performance comparison on CV tasks.

|  | ViT-B/32 | | | ViT-B/16 | | |
| --- | --- | --- | --- | --- | --- | --- |
|  | ACC | Unify Time (s) | Inference Speed (samples/s) | ACC | Unify Time (s) | Inference Speed (samples/s) |
| BTM | 90.33 | – | 606 | 91.75 | – | 249 |
| ModelSoup | 74.20 | 5.72 | 1813 | 79.42 | 5.72 | 743 |
| BTX | 74.30 | 11.13 | 1758 | 81.20 | 19.72 | 715 |
| FlexOlmo | 92.92 | 11.93 | 1767 | 93.53 | 18.24 | 719 |
| HarmoMoE | **94.52** | 12.15 | 1751 | **96.24** | 19.88 | 710 |

Table 16: Cost–performance comparison on NLP tasks.

|  | LLaMA-3.2-3B | | | LLaMA-3.1-8B | | |
| --- | --- | --- | --- | --- | --- | --- |
|  | ACC | Unify Time (s) | Inference Speed (samples/s) | ACC | Unify Time (s) | Inference Speed (samples/s) |
| BTM | 72.74 | – | 17.44 | 79.79 | – | 8.20 |
| ModelSoup | 73.57 | 8.24 | 43.46 | 80.51 | 11.25 | 22.86 |
| BTX | 71.14 | 118.21 | 42.12 | 76.73 | 223.37 | 21.41 |
| FlexOlmo | 72.50 | 119.90 | 41.59 | 77.46 | 206.28 | 21.95 |
| HarmoMoE | **74.42** | 114.46 | 40.67 | **81.59** | 205.42 | 20.05 |

