# OpenReview forum: "HarmoMoE: Unifying Domain-Specialized Experts into a Mixture-of-Experts Model under Privacy Constraints"
_ICLR.cc/2026/Conference — Submitted to ICLR 2026_

### Official Review · Reviewer_VQAo · 2025-10-29

**Soundness:** 3
**Presentation:** 3
**Contribution:** 3
**Rating:** 6
**Confidence:** 3

**Summary:**

This paper proposes a framework for enabling data-distributed training without sacrificing data owners’ privacy. Their approach is to train diverse experts using local data for each client, and then construct proxy samples for fine-tuning private models and router learning. The method has shown superior performance over SOTA baselines.

**Strengths:**

- This paper provides a novel method to merge domain-specific experts into a versatile expert, with a special focus on data privacy. The method is promising.

- The authors introduce several key elements, compared to the existing baselines: DPP proxy data sampling, context-aware router, and proxy-aligned expert training. Each component is ablated rigorously and is proven empirically to have contributed to the performance improvement.

**Weaknesses:**

- An immediate dropback is that what if there are no similar data to $D_p$ in the public dataset to construct $\hat{D}_p$? Does the procedure still work?

- There is no documentation of the computational cost in addition to BTM. Is the enhanced performance coming at a greater cost?

**Questions:**

- In Table 3, it seems the domain-specific experts are not performing the best in the corresponding domains?

- There are several other works with similar purposes. How does your method compare to [1][2]?

- To get a complete understanding of the method, could you please replicate the experiment from Section 4.4 on the vision tasks and the experiment from Section 4.5 on the NLP tasks?


[1] On-Device Collaborative Language Modeling via a Mixture of Generalists and Specialists

[2] Mixture-of-LoRAs: An Efficient Multitask Tuning for Large Language Models

---

> ### Author Response · Authors · 2025-11-20
> **Reply to Reviewer VQAo (Part 1)**
>
> Thank you for the positive rating and valuable feedback. We address your concerns as follows.
>
> ---
>
> > **Q1.** An immediate dropback is that what if there are no similar data to $\mathcal{D}_p$ in the public dataset $\hat{\mathcal{D}}_p$ to construct? Does the procedure still work?
>
> **A1.**
> Thank you for raising this question.
> The HarmoMoE procedure still works even when no similar public data exist, because HarmoMoE (i) injects domain cues into the router via expert-derived initialization, (ii) uses proxies only as coordination anchors, and (iii) calibrates each expert on its own proxies to preserve router–expert compatibility.
>
> **(1) Router initialization provides domain cues.**
> The router is initialized using expert-derived embeddings computed on both client and proxy data (Lines 268-272, Steps 11 and 15 in Algorithm 1), which encode each expert’s domain characteristics while preserving privacy. **This initialization injects domain-aware cues into the router** and enables effective expert coordination even when the public data contain no directly related content.
>
> **(2) Proxy data can improve experts coordination without requiring close similarity between proxy and client data.**
> Proxy samples are used solely for router training to coordinate experts. While higher similarity between proxies and private data can further improve expert coordination, HarmoMoE does not heavily rely on such similarity—proxies can be only **conceptually related** and still serve as effective coordination anchors that let the router learn relative expert behaviors. In practice, **modern public corpora (e.g., ImageNet-21K with 14M images, Common Crawl with 150 to 350 Billion tokens) are sufficiently broad and diverse that this mild assumption of having some loosely related data is easily satisfied.**
>
> **(3) Proxy-aligned expert training ensures router–expert compatibility under distribution gaps.**
> Each expert is finetuned on both its private and proxy data, ensuring consistent behavior on **the same proxy distribution later used for router learning**. This alignment maintains expert–router compatibility even when the client and public data distributions differ substantially.
>
> Empirically, our NLP experiments already instantiate this scenario: Alpaca is a general-purpose instruction dataset **with no domain overlap with CommonsenseQA, CosmosQA, or SocialIQA**, HarmoMoE **successfully unifies these clients into a single MoE model and achieves strong performance** (Tables 3 and 4 in our paper).

---

> ### Author Response · Authors · 2025-11-20
> **Reply to Reviewer VQAo (Part 2)**
>
> > **Q2.** There is no documentation of the computational cost in addition to BTM. Is the enhanced performance coming at a greater cost?
>
> **A2.**
> Thank you for the question. In the tables below, we provide a detailed comparison of HarmoMoE’s computational cost against existing methods, covering both **unification time** and **inference speed**. The results show that HarmoMoE achieves strong accuracy improvements without adding significant overhead on either vision or language tasks.
> Specifically, we can observe that:
>
> **(1) Unification cost: nearly identical to existing MoE methods.**
> The merging (unification) time of HarmoMoE is nearly identical to FlexOlmo and BTX across all backbones, showing that the accuracy gains do not come from higher computational cost. While BTM and ModelSoup appear faster (or cost-free), they avoid the coordination required for MoE merging, which explains their lower accuracy. The small extra cost in HarmoMoE is therefore a modest and worthwhile trade-off for its significant accuracy improvement.
>
> **(2) Inference efficiency: comparable to other MoE methods.**
> HarmoMoE maintains inference speeds comparable to other MoE unification methods, confirming that the context-aware router introduces minimal runtime overhead and scales efficiently across backbones.
> In contrast, **BTM requires inference over all experts for every input**, leading to **2x slower inference** despite its zero unification cost.
>
> **(3) Overall cost–performance balance.**
> Across both CV and NLP tasks, HarmoMoE consistently achieves **state-of-the-art accuracy with comparable computational efficiency**, as summarized below.
>
> `Table: Cost–Performance Comparison on CV Tasks`
>
> $$
> \begin{array}{l|ccc|ccc}
> \hline
> & & \text{ViT-B/32} & && \text{ViT-B/16} \newline
> \hline
> & \text{ACC}&\text{Unify Time}&\text{Inference Speed} & \text{ACC}&\text{Unify Time}&\text{Inference Speed} \newline
> &&  \text{(seconds)} & \text{(samples/second)} & &\text{(seconds)} & \text{(samples/second)} \newline
> \hline
> \text{BTM} & 90.33&-&606 & 91.75&-&249 \newline
> \text{ModelSoup} & 74.20 & 5.72 & 1813 & 79.42 & 5.72 & 743 \newline
> \text{BTX} & 74.30 & 11.13 & 1758 & 81.20 & 19.72 & 715 \newline
> \text{FlexOlmo} & 92.92 & 11.93 & 1767 & 93.53 & 18.24 & 719 \newline
> \text{HarmoMoE} & \textbf{94.52} & 12.15 & 1751 & \textbf{96.24} & 19.88 & 710 \newline
> \hline
> \end{array}
> $$
>
> `Table: Cost–Performance Comparison on NLP Tasks`
>
> $$
> \begin{array}{l|ccc|ccc}
> \hline
> &  &\text{LLaMA-3.2-3B} & &&\text{LLaMA-3.1-8B}& \newline
> \hline
> & \text{ACC}  & \text{Unify Time} & \text{Inference Speed} & \text{ACC} & \text{Unify Time} & \text{Inference Speed} \newline
> &&  \text{(seconds)} & \text{(samples/second)} & &\text{(seconds)} & \text{(samples/second)} \newline
> \hline
> \text{BTM} & 72.74 & - & 17.44 & 79.79 & - & 8.20 \newline
> \text{ModelSoup} & 73.57 & 8.24 & 43.46 & 80.51 & 11.25 & 22.86 \newline
> \text{BTX} & 71.14 & 118.21 & 42.12 & 76.73 & 223.37 & 21.41 \newline
> \text{FlexOlmo} & 72.50 & 119.90 & 41.59 & 77.46 & 206.28 & 21.95 \newline
> \text{HarmoMoE} & \textbf{74.42} & 114.46 & 40.67 & \textbf{81.59} & 205.42 & 20.05 \newline
> \hline
> \end{array}
> $$
>
> We have included these discussions in Appendix H.7 of the revised paper.
>
> ---
>
> > **Q3.** In Table 3, it seems the domain-specific experts are not performing the best in the corresponding domains?
>
> **A3.**
> We thank the reviewer for spotting this inconsistency. It was a typo in the original manuscript where the performance rows for Expert II (CosmosQA) and Expert III (SocialIQA) were switched.
> As shown in the corrected data, the domain-specific experts do achieve the highest performance on their specific tasks, confirming that the experts are effectively specialized for their target tasks.
>
> $$
> \begin{array}{lcccc}
> \hline
>  & \text{CommonsenseQA} & \text{CosmosQA} & \text{SocialIQA} & \textbf{Average} \newline
> \hline
> \text{UnrestrictedMoE} & 75.51 & 78.39 & 71.80 & 75.23 \newline
> \hline
> \text{ZeroShot} & 62.49 & 62.68 & 56.19 & 60.45 \newline
> \text{Expert I (CommonsenseQA)} & 74.94 & 70.18 & 60.54 & 68.55 \newline
> \text{Expert II (CosmosQA)} & \color{red}{63.06} & \color{red}{78.69} & \color{red}{58.96} & \color{red}{66.90} \newline
> \text{Expert III (SocialIQA)} & \color{red}{65.44} & \color{red}{68.04} & \color{red}{72.42} & \color{red}{68.63} \newline
> \hline
> \text{BTM} & 74.61 & 75.44 & 68.17 & 72.74 \newline
> \text{ModelSoup} & 73.71 & 75.24 & 71.75 & 73.57 \newline
> \text{BTX} & 69.62 & 72.06 & 71.75 & 71.14 \newline
> \text{FlexOlmo} & 73.30 & 73.33 & 70.88 & 72.50 \newline
> \text{HarmoMoE} & \textbf{74.94} & \textbf{76.05} & \textbf{72.26} & \textbf{74.42} \newline
> \hline
> \end{array}
> $$
>
> We have fixed this typo in Table 3 of the revised paper.

---

> ### Author Response · Authors · 2025-11-20
> **Reply to Reviewer VQAo (Part 3)**
>
> > **Q4.** There are several other works with similar purposes. How does your method compare to [1][2]?
> >
> > [1] On-Device Collaborative Language Modeling via a Mixture of Generalists and Specialists
> >
> > [2] Mixture-of-LoRAs: An Efficient Multitask Tuning for Large Language Models
>
> **A4.**
> Thank you for pointing out these related works.
> While CoMiGS [1] and MoA [2] both pursue multi-expert unification, HarmoMoE differs fundamentally in its training paradigm and privacy assumption. Below we discuss these distinctions and present direct empirical comparisons.
>
> **(1) Comparison with CoMiGS.**
> CoMiGS adopts a **federated learning (FL)** paradigm that requires periodically exchanging model parameters for joint optimization. This approach incurs **high communication and memory costs** and often becomes unstable under heterogeneous client data, leading to **degraded performance**.
> In contrast, **HarmoMoE avoids exchanging model parameters periodically**. Each client independently fine-tunes its expert on private and proxy data, and all experts are unified once through MoE integration. This merge-after-training design achieves better scalability, eliminates communication overhead, and remains stable under heterogeneous data.
>
> **(2) Comparison with Mixture-of-LoRAs (MoA).**
>  MoA also seeks to unify multiple LoRA experts but **assumes direct access to client private data for router training** (mentioned in section 3.1 of their paper), which violates privacy constraints.
> In our additional experiments, we reimplemented MoA by training its router only on public data to maintaining expert coordination without violating privacy constraints.
>
> **(3) Empirical results on NLP tasks.**
> We evaluated CoMiGS and MoA under our setting using the same LoRA configurations.
> Due to memory constraints, **CoMiGS could not scale to LLaMA-3.1-8B (out-of-memory issues)**.
> As shown in the tables below, **HarmoMoE consistently outperforms both methods across all benchmarks**.
>
> `Table: Results on NLP Tasks (LLaMA-3.2-3B)`
> $$
> \begin{array}{l|cccc}
> \hline
> & \text{CommonsenseQA} & \text{CosmosQA} & \text{SocialIQA} & \textbf{Average} \newline
> \hline
> \text{CoMiGS [1]}  &  72.32 & 71.46 & 71.19 & 71.65  \newline
> \text{MoA [2]}  & 71.09 & 74.64 & 70.11 & 71.95  \newline
> \text{HarmoMoE}& \textbf{74.94} & \textbf{76.05} & \textbf{72.26} & \textbf{74.42} \newline
> \hline
> \end{array}
> $$
>
> `Table: Results on NLP Tasks (LLaMA-3.1-8B)`
> $$
> \begin{array}{l|cccc}
> \hline
> & \text{CommonsenseQA} & \text{CosmosQA} & \text{SocialIQA} & \textbf{Average} \newline
> \hline
> \text{CoMiGS [1]}  &  - & - & - & - \newline
> \text{MoA [2]}  &  77.72 & 81.64 &74.05 & 77.81  \newline
> \text{HarmoMoE}& \textbf{81.33} & \textbf{85.80} & \textbf{77.64} & \textbf{81.59} \newline
> \hline
> \end{array}
> $$
>
> We have included these comparisons in Appendix H.2 of the revised paper.
>
> ---
>
> > **Q5.** To get a complete understanding of the method, could you please replicate the experiment from Section 4.4 on the vision tasks and the experiment from Section 4.5 on the NLP tasks?
>
> **A5.** We appreciate the reviewer's constructive suggestion to broaden our ablation studies. To provide a more complete understanding of our method's generality, we have conducted the experiments from Section 4.4 (Context-aware Router Ablation) on the CV tasks and Section 4.5 (Proxy-Aligned Expert Training Ablation) on the NLP tasks.
> The results, detailed below, confirm that **both the context-aware router and proxy-aligned expert training provide consistent benefits across both NLP and CV tasks**.
>
> `Table: Results on CV Tasks (Context-Aware Router)`
>
> $$
> \begin{array}{l|c|cccc}
> \hline
>  & \text{Context-Aware Router} & \text{Pets} & \text{Flowers} & \text{EuroSAT} & \textbf{Average} \newline
> \hline
> \text{ViT-B/32} & \textsf{x} & 91.69 & 93.18 & 96.90 & 93.92 \newline
> \text{ViT-B/32} & \checkmark & \mathbf{91.91} & \mathbf{93.67} & \mathbf{97.98} & \mathbf{94.52} \newline
> \hline
> \text{ViT-B/16} & \textsf{x} & 93.95 & 96.71 & 97.16 & 95.94 \newline
> \text{ViT-B/16} & \checkmark & \mathbf{94.22} & \mathbf{97.08} & \mathbf{97.41} & \mathbf{96.24} \newline
> \hline
> \end{array}
> $$
>
> `Table: Results on NLP Tasks (Proxy-Aligned Expert Training)`
>
> $$
> \begin{array}{l|c|cccc}
> \hline
>  & \text{Proxy-Aligned Training} & \text{CommonsenseQA} & \text{CosmosQA} & \text{SocialIQA} & \textbf{Average} \newline
> \hline
> \text{LLaMA-3.2-3B} & \textsf{x} & 74.77 & 71.56 & 71.80 & 72.71 \newline
> \text{LLaMA-3.2-3B} & \checkmark & \mathbf{74.94} & \mathbf{76.05} & \mathbf{72.26} & \mathbf{74.42} \newline
> \hline
> \text{LLaMA-3.1-8B} & \textsf{x} & 80.75 & 84.89 & 77.33 & 80.99 \newline
> \text{LLaMA-3.1-8B} & \checkmark & \mathbf{81.33} & \mathbf{85.80} & \mathbf{77.64} & \mathbf{81.59} \newline
> \hline
> \end{array}
> $$
>
> We have revised the paper to include the extended ablation studies in Sections 4.4 and 4.5.

---

> > ### Comment · Reviewer_VQAo · 2025-11-26
> >
> > Thanks for the detailed response.
> >
> > I have some further questions regarding the baseline implementation:
> >
> > 1) From the newly added tables, BTX appears to consistently underperform BTM, even though BTX is built on top of BTM and is reported in [1] to achieve better results. This raises some concerns about the implementation of the baseline methods.
> >
> > 2) I reviewed the CoMiGS paper, and it highlights the capability to work effectively with heterogeneous client data by single out some experts for personalization, which is different than what you claimed. In addition, why would it run out of memory? It also relies on a set of LoRA expert modules, and sharing a single expert shouldn’t, in principle, lead to memory overflow.
> >
> > 3) For a fair comparison with Mixture-of-LoRAs, wouldn’t it be more appropriate to train the router using the proxy dataset $\hat{D}_p$ rather than the full public dataset?
> >
> > [1] Branch-Train-MiX: Mixing Expert LLMs into a Mixture-of-Experts LLM

---

> > > ### Author Response · Authors · 2025-11-28
> > > **Reply to Reviewer VQAo (Part 4)**
> > >
> > > Thanks for the further comments.
> > > We address your concerns as follows.
> > >
> > > > **Q6.** From the newly added tables, BTX appears to consistently underperform BTM, even though BTX is built on top of BTM and is reported in [1] to achieve better results. This raises some concerns about the implementation of the baseline methods.
> > >
> > > **A6.**
> > > Thank you for raising this concern. We confirm that our **BTX implementation is correct and faithfully follows the original design [1]**, but its lower performance compared to BTM arises from differences in the **privacy-preserving setting** rather than implementation issues. Below, we clarify the reasons.
> > >
> > > **(1) Why BTX underperforms BTM in our setting.**
> > > BTM and BTX adopt fundamentally different unification strategies. **BTM** performs **output-level ensembling**, directly combining expert predictions without parameter merging, whereas **BTX** performs **parameter-level merging** into a single MoE model with a learned router.
> > > In the BTX paper, training the router relies on **client private data** to achieve accurate domain alignment. However, accessing client private data is **infeasible under our privacy-preserving constraints**. For fairness, we train BTX using only **public proxy data**, which weakens its domain alignment and thus reduces performance relative to BTM.
> > >
> > > **(2) Why HarmoMoE outperforms both BTX and BTM.**
> > > Our HarmoMoE incorporates **three key improvements** that mitigate the limitations of BTX:
> > > * **Relevance-weighted DPP** selects proxies that are both diverse and highly aligned with each client’s domain.
> > > * **Proxy-aligned expert training** fine-tunes each expert jointly on private and proxy data, improving domain consistency before unification.
> > > * **Context-aware routing** leverages global input context for more reliable expert selection during inference.
> > >
> > > These enhancements enable HarmoMoE to achieve stronger alignment and generalization than both BTM and BTX while maintaining full privacy preservation.
> > >
> > > ---
> > >
> > > > **Q7.** I reviewed the CoMiGS paper, and it highlights the capability to work effectively with heterogeneous client data by single out some experts for personalization, which is different than what you claimed.
> > >
> > > **A7.**
> > > Thank you for the comment. We agree that **CoMiGS** is designed to handle heterogeneous client data by assigning certain experts for personalization. However, this approach and objective fundamentally differ from **HarmoMoE**, which focuses on building a **single unified MoE model** that generalizes across all domains. Below, we clarify the distinction and explain why HarmoMoE achieves stronger performance.
> > >
> > > **(1) CoMiGS performs federated personalization, not unified generalization.**
> > > In CoMiGS, each of the $K$ clients trains its own MoE model consisting of **generalist** and **specialist** experts. During aggregation, only the generalist experts are exchanged and averaged, while the specialist experts remain local to each client for personalization. Consequently, **CoMiGS produces $K$ personalized MoEs**, each specialized to its own client distribution, rather than one unified model applicable to all domains.
> > >
> > > **(2) Why HarmoMoE outperforms CoMiGS.**
> > > In CoMiGS, each client maintains its own **specialist experts** that are fine-tuned specifically for its private domain. At inference time, the framework must therefore know **which client (or task) a given input belongs to** in order to activate the corresponding specialist experts. This requires access to a **client identifier**, which is infeasible in our **domain-agnostic** setting (i.e., the model does not know the source of each input). Consequently, we evaluate CoMiGS using only its **generalist experts**, as its personalization mechanism cannot operate without client identity information.
> > >
> > > In contrast, **HarmoMoE** unifies all client experts into a single deployable MoE and learns to **route inputs automatically based on their content**, without any client or task identifiers. Its superior performance arises from three key designs—**relevance-weighted DPP** for proxy selection, **proxy-aligned expert training**, and **context-aware routing**—which together enable strong cross-domain generalization under privacy constraints **without relying on client identity information**.
> > >
> > > **PS:** If there exists a way to utilize CoMiGS’s specialist experts without requiring domain or client identifiers, we would greatly appreciate the reviewer’s guidance and are happy to implement and evaluate it for completeness.

---

> ### Author Response · Authors · 2025-11-28
> **Reply to Reviewer VQAo (Part 5)**
>
> > **Q8.** In addition, why would it run out of memory? It also relies on a set of LoRA expert modules, and sharing a single expert shouldn’t, in principle, lead to memory overflow.
>
> **A8.**
> Thank you for the question. Although CoMiGS employs **LoRA adapters**, the OOM issue arises from its **federated learning (FL) training design**, not from the expert type itself. In CoMiGS, **each client maintains its own copy of the backbone model** together with its **generalist and specialist LoRA experts**. During synchronous FL training, all $K$ client models (each with a full backbone and associated LoRA modules) must be loaded **simultaneously** to perform gradient updates and aggregation. As a result, GPU memory consumption grows roughly linearly with the number of clients, leading to OOM when training larger models (e.g., LLaMA-3.1-8B used in our experiments).
>
> This limitation is **not specific to CoMiGS**, but a **common challenge for FL-based methods**, where multi-client synchronization becomes infeasible as model size increases. The CoMiGS paper itself confines experiments to small models (GPT-2-124M and Llama-3.2-1B) to keep this training scheme tractable.
>
> In principle, **training clients on separate servers** could mitigate the OOM issue, as each client process would have dedicated hardware resources. However, we adopted the **official CoMiGS implementation**, which trains all $K$ clients on the **same server**, leading to the observed OOM issue when scaling to larger models.
>
> In contrast, **HarmoMoE completely avoids this limitation**. Each client expert is trained **independently and asynchronously** on its private and proxy data, without concurrent model loading or synchronized updates. The experts are merged only once during the MoE integration step. This asynchronous, communication-free design makes HarmoMoE **memory-efficient, scalable, and practical for large-model unification** under privacy-preserving constraints.
>
> ---
>
> > **Q9.** For a fair comparison with Mixture-of-LoRAs, wouldn’t it be more appropriate to train the router using the proxy dataset $\hat{\mathcal{D}}_p$ rather than the full public dataset?
>
> **A9.**
> Thank you for the suggestion. We clarify that we trained the router on the **full public dataset** rather than the proxy dataset $\hat{\mathcal{D}}_p$ to ensure a consistent comparison with **Mixture-of-LoRAs (MoA)**, as the original MoA framework does **not** include any proxy-selection mechanism. Using the full public dataset therefore follows MoA’s original design and isolates the benefit of our **relevance-weighted proxy selection**.
>
> For completeness, we also trained MoA with its router learned on $\hat{\mathcal{D}}_p$. As shown in the tables below, this version (MoA w/ $\hat{\mathcal{D}}_p$) performs better than MoA trained on the full public dataset (MoA w/o $\hat{\mathcal{D}}_p$), confirming that **proxy data is indeed beneficial** and validating the usefulness of our **relevance-weighted proxy selection**.
> Moreover, **HarmoMoE consistently outperforms MoA w/ $\hat{\mathcal{D}}_p$**, demonstrating the additional advantages brought by our proposed **proxy-aligned expert training** and **context-aware routing**. Together, these components enable HarmoMoE to achieve stronger cross-domain generalization under privacy constraints.
>
>
> `Table: Results on NLP Tasks (LLaMA-3.2-3B)`
> $$
> \begin{array}{l|cccc}
> \hline
> & \text{CommonsenseQA} & \text{CosmosQA} & \text{SocialIQA} & \textbf{Average} \newline
> \hline
> \text{MoA w/o }  \hat{\mathcal{D}_p}  & 71.09 & 74.64 & 70.11 & 71.95  \newline
> \text{MoA w/ }  \hat{\mathcal{D}_p}  &  72.21 & 74.84 & 70.91 & 72.65 \newline
> \text{HarmoMoE}& \textbf{74.94} & \textbf{76.05} & \textbf{72.26} & \textbf{74.42} \newline
> \hline
> \end{array}
> $$
>
> `Table: Results on NLP Tasks (LLaMA-3.1-8B)`
> $$
> \begin{array}{l|cccc}
> \hline
> & \text{CommonsenseQA} & \text{CosmosQA} & \text{SocialIQA} & \textbf{Average} \newline
> \hline
> \text{MoA w/o }  \hat{\mathcal{D}_p}  &  77.72 & 81.64 &74.05 & 77.81  \newline
> \text{MoA w/ }  \hat{\mathcal{D}_p}  &  78.05 & 82.45 & 75.18 & 78.56  \newline
> \text{HarmoMoE}& \textbf{81.33} & \textbf{85.80} & \textbf{77.64} & \textbf{81.59} \newline
> \hline
> \end{array}
> $$

---

### Official Review · Reviewer_ZFYq · 2025-10-30

**Soundness:** 3
**Presentation:** 3
**Contribution:** 3
**Rating:** 6
**Confidence:** 3

**Summary:**

The manuscript proposes HarmoMoE, a framework designed to unify expert models trained on private data without requiring coordinated retraining. HarmoMoE uses a relevance-based determinantal point process to select diversified and domain-representative proxy samples, allowing the router to be trained in a harmonized manner using abundant approximate data. A context-aware router further refines the overall design.

**Strengths:**

1. The baseline selection is up to date.
2. The motivation for unifying expert models is well explained.

**Weaknesses:**

1. The main concern lies in the privacy-preserving claim. HarmoMoE uses relevance-weighted DPP to select proxy data that represent private data. However, if the proxy data are highly similar to the private data, wouldn’t this constitute a form of data exposure? If not, how does this differ from a vanilla DPP? I suggest adding more discussion about the privacy–utility trade-off involved in using such public proxy data.
2. HarmoMoE focuses on unifying full-rank experts, but extending the approach to low-rank adapters seems both more feasible and practical in many real-world settings.

**Questions:**

1. Given the assumption that D_0 contains sufficient public data that are representative of private client data, why not simply allow the cloud to train directly on the entire D_0? This baseline should be included to highlight the unique effectiveness of HarmoMoE.
2. Please discuss the relationship between HarmoMoE and low-rank adaptation unification methods (e.g., LoRASuite, NeurIPS 2025). While additional experiments are not mandatory, even a small-scale or illustrative experiment could strengthen the empirical validation.
3. What is the proportion of public versus private data used in the experiments?

---

> ### Author Response · Authors · 2025-11-20
> **Reply to Reviewer ZFYq (Part 1)**
>
> Thank you for your positive rating and valuable feedback. We address your concerns as follows.
>
> ---
>
> > Q1. The main concern lies in the privacy-preserving claim. HarmoMoE uses relevance-weighted DPP to select proxy data that represent private data. However, if the proxy data are highly similar to the private data, wouldn’t this constitute a form of data exposure?
>
> **A1.**
> Thank you for the insightful question. We clarify that **HarmoMoE never exposes private data**, and any “exposure” that occurs during proxy selection pertains **only to public data**, not to client data. Therefore, it does **not** violate privacy or data-residency requirements.
>
> **(1) Semantic similarity is not equivalent to private-data exposure.**
> Our relevance-weighted DPP method selects public samples that are representation-wise similar to the client domain. These proxies may resemble private data but are not derived from private samples, and all clients already have access to them.
> In privacy-preserving systems, leakage occurs only when **non-public information becomes newly revealed**. Selecting an already public sample—no matter how similar—does not expose any new private information.
>
> **(2) Overlap with public data does not constitute private-data exposure.**
> If a public sample coincidentally overlaps with one in a client’s dataset, revealing that sample still constitutes **public-data exposure**, not private-data exposure, since the content is already publicly available prior to any interaction with HarmoMoE.
> This principle aligns with standard privacy frameworks such as the **California Consumer Privacy Act (CCPA, https://en.wikipedia.org/wiki/California_Consumer_Privacy_Act)**, which explicitly excludes “publicly available information” from the definition of personal data. Under this definition, **the exposure of a public sample is not regarded as a violation of privacy.**
>
> **(3) HarmoMoE introduces no new channels for private-data leakage.**
> The proxy-selection process transmits only the IDs of selected public samples and the final expert weights—never private samples, gradients, or intermediate activations. Because all proxy candidates are drawn from a public dataset, the process does not disclose or allow inference about private data.
>
> We have included this discussion in Appendix H.5 of the revised paper.
>
> ---
>
> > **Q2.** If not, how does this differ from a vanilla DPP?
>
> **A2.** Our approach differs from a vanilla DPP by explicitly incorporating a **relevance term** to handle the distribution shift between public and private data.
> - If all public proxy data are highly similar to the client’s private distribution, our method would naturally reduce to a vanilla (diversity-only) DPP.
>
> - **However, in practice, this is rarely the case.** Public datasets (e.g., ImageNet) are typically broad and generic, while client data is often specialized and distinct. In this realistic scenario, a vanilla DPP is insufficient because it blindly maximizes diversity, often selecting diverse samples that are semantically irrelevant to the client's specific task.
>
> Our **relevance-weighted DPP** addresses this by enforcing two criteria simultaneously:
> **(i)** **Relevance**, ensuring selected proxies are semantically aligned with the client data, and
> **(ii)** **Diversity**, preventing redundancy among the selected proxy samples.
>
> We validated this empirically by comparing our method against a vanilla DPP on both CV tasks (ViT-B/32) and NLP tasks (LLaMA-3.2-3B). As shown below, **incorporating relevance yields consistent performance gains** (+2.72% in Vision and +1.78% in NLP), confirming that diversity alone is insufficient for effective proxy selection.
>
> `Table: Results on Vision Tasks (ViT-B/32)`
> $$
> \begin{array}{lcccc}
> \hline
>  & \text{Pets} & \text{Flowers} & \text{EuroSAT} & \textbf{Average} \newline
> \hline
> \text{DPP-only} & 90.81 & 91.00 & 93.59 & 91.80 \newline
> \text{Relevance-weighted DPP} & \textbf{91.91} & \textbf{93.67} & \textbf{97.98} & \textbf{94.52} \newline
> \hline
> \end{array}
> $$
>
> `Table: Results on NLP Tasks (LLaMA-3.2-3B)`
> $$
> \begin{array}{lcccc}
> \hline
>  & \text{C-QA} & \text{Cosmos} & \text{Social} & \textbf{Average} \newline
> \hline
> \text{DPP-only} & 73.38 & 74.04 & 70.51 & 72.64 \newline
> \text{Relevance-weighted DPP} & \textbf{74.94} & \textbf{76.05} & \textbf{72.26} & \textbf{74.42} \newline
> \hline
> \end{array}
> $$
>
> ---
>
> > **Q3.** I suggest adding more discussion about the privacy–utility trade-off involved in using such public proxy data.
>
> **A3.**
> Thank you for the suggestion. We respectfully clarify that **HarmoMoE preserves privacy entirely, so there is no privacy–utility trade-off**. All proxies are selected solely from public data, and no private samples are ever shared. The relevance-weighted DPP only refines how closely public proxies align with private domains, thereby enhancing utility without compromising privacy.

---

> ### Author Response · Authors · 2025-11-20
> **Reply to Reviewer ZFYq (Part 2)**
>
> > **Q4.** HarmoMoE focuses on unifying full-rank experts, but extending the approach to low-rank adapters seems both more feasible and practical in many real-world settings.
>
> **A4.**
> Thank you for the suggestion.
> Our proposed HarmoMoE **supports both full-rank and low-rank experts** and is not limited to any specific expert type. In practice, **our implementation is already low-rank and parameter-efficient**. All HarmoMoE experiments employ LoRA experts (Lines 297-300, 370-373), where each expert is a low-rank LoRA adapter built on a frozen backbone, and client specialization updates only these LoRA parameters.
>
> ---
>
> > **Q5.** Given the assumption that D_0 contains sufficient public data that are representative of private client data, why not simply allow the cloud to train directly on the entire D_0? This baseline should be included to highlight the unique effectiveness of HarmoMoE.
>
> **A5.**
> Thank you for the suggestion.
> We clarify that **training directly on the full public dataset $\mathcal{D}_0$ is ineffective**. Although $\mathcal{D}_0$ may include samples resembling private data, it is dominated by unrelated content and thus **fails to represent client-specific distributions**, resulting in poor specialization.
>
> To verify this, we conducted a CV experiment with CLIP ViT-B/32 to compare a model trained directly on $\mathcal{D}_0$ with HarmoMoE. As shown below, training on $\mathcal{D}_0$ achieves only **49.80%** average accuracy, while HarmoMoE reaches **94.52%**, a **+44.7%** improvement.
> These results confirm that **HarmoMoE’s strength lies in unifying domain-specialized experts through relevance-diverse proxy selection**, achieving coordinated expertise that simple training on $\mathcal{D}_0$ cannot replicate.
>
> $$
> \begin{array}{l|cccc}
> \hline
> & \text{Pets} & \text{Flowers} & \text{EuroSAT} & \textbf{Average} \newline
> \hline
> \text{Train on }\mathcal{D}_0 & 81.17 & 41.33 & 26.91 & 49.80 \newline
> \text{HarmoMoE} & \mathbf{91.91} & \mathbf{93.67} & \mathbf{97.98} & \mathbf{94.52} \newline
> \hline
> \end{array}
> $$
>
> ---
>
> > **Q6.** Please discuss the relationship between HarmoMoE and low-rank adaptation unification methods (e.g., LoRASuite, NeurIPS 2025). While additional experiments are not mandatory, even a small-scale or illustrative experiment could strengthen the empirical validation.
>
> **A6.**
> Thank you for the question and for pointing us to LoRASuite. We clarify that **HarmoMoE is a low-rank adaptation unification method**, whereas **LoRASuite is not**, and the two address different goals.
>
> **(1) Different goals: LoRA unification vs. LoRA migration.**
> HarmoMoE aims to **unify multiple domain-specialized LoRA experts** trained on the same backbone into a privacy-preserving MoE—answering how to combine many LoRA experts without sharing client data.
> LoRASuite instead focuses on **LoRA migration**, transferring a single LoRA adapter trained on backbone (A) to backbone (B) after the backbone is upgraded. Its goal is expert adaptation, not expert unification.
>
> **(2) Different data assumptions.**
> LoRASuite requires **client private data** to align activations and performs poorly without it, while HarmoMoE assumes **no private data access** and relies entirely on public proxies for supervision.
>
> **(3) Why a direct comparison is inappropriate.**
> A direct empirical comparison would be inappropriate, as it would force LoRASuite into a multi-expert unification setting it was never designed for.
>
> We have included this discussion in Appendix H.6 of the revised paper.
>
> ---
>
> > **Q7.** What is the proportion of public versus private data used in the experiments?
>
> **A7.**
> In all experiments, **only a small portion of public data is used**, as the lightweight router requires minimal supervision to learn effective expert coordination. Specifically, the proportion of selected public proxies to private data is 16.98%, 12.22%, and 3.70% for the vision tasks (Pets, Flowers, EuroSAT) and 5.13%, 1.98%, and 1.50% for the NLP tasks (CommonsenseQA, CosmosQA, SocialIQA).
>
> These ratios correspond to a fixed budget of 500 public proxy samples per client. This modest amount is sufficient because the context-aware router is extremely lightweight—comprising only 27,648 parameters for the CV experiments and 258,048 / 393,216 for the NLP models (LLaMA-3.2-3B and LLaMA-3.1-8B). The router’s compact design ensures strong performance without relying on large-scale public data.
>
> For reference, the private training set sizes are:
> * CV: Pets (2,944), Flowers (4,093), EuroSAT (13,500)
> * NLP: CommonsenseQA (9,741), CosmosQA (25,262), SocialIQA (33,410)

---

### Official Review · Reviewer_gTvc · 2025-10-31

**Soundness:** 3
**Presentation:** 2
**Contribution:** 2
**Rating:** 4
**Confidence:** 3

**Summary:**

This paper proposes a privacy-preserving training method for sparse Mixture-of-Experts models. To unify multiple expert models, each trained on separate private data, into a single MoE model, the paper proposes a proxy-data selection strategy (weighted DPP) and a context-aware router-training strategy to train the router within the unified model. The empirical results presented in the paper demonstrate that the proposed method outperforms previous baselines in privacy-preserving unification to MoE models.

**Strengths:**

1. The proposed method is logically designed

2. The empirical results outperform previous baselines

3. The paper is well-structured and easy to follow

**Weaknesses:**

1. **Underperformance**: The proposed method outperforms previous privacy-preserving unification baselines. But it still underperforms compared to separately finetuned models on private data. My concern is that, if the proposed unification method does not improve results after unification, what is the advantage of unification? In that case, each client can use their respective finetuned model and enjoy better performance.

2. **Potential suboptimal design**: The paper incorporated the context-aware router training, where the input tokens contain a component average over all the tokens of the input sequence, to capture the input context. Although the design choice improves performance over router training without the context-aware component, the design choice may be suboptimal.

**Questions:**

1. Can the authors explain why the proposed technique is advantageous despite having lower performance than the individually trained models on private data?

2. Can the authors discuss why the proposed context-aware design is optimal?

---

> ### Author Response · Authors · 2025-11-20
> **Reply to Reviewer gTvc (Part 1)**
>
> Thank you for the insightful review. We address your concerns as follows.
>
> ---
>
> > **Q1.** Underperformance: The proposed method outperforms previous privacy-preserving unification baselines. But it still underperforms compared to separately finetuned models on private data. My concern is that, if the proposed unification method does not improve results after unification, what is the advantage of unification? In that case, each client can use their respective finetuned model and enjoy better performance.
>
> > Can the authors explain why the proposed technique is advantageous despite having lower performance than the individually trained models on private data?
>
> **A1.**
> Thank you for the question.
> The primary goal of **HarmoMoE** is to construct a **single, privacy-preserving Mixture-of-Experts (MoE)** that every client can use **without sharing private data**. It is not designed to outperform each client’s individual expert on its **own** domain, but rather to enable all clients to benefit from a **multi-domain, unified MoE model** that none could train alone under privacy constraints.
>
> **(1) Unification provides each client with a stronger, shared model instead of remaining isolated.**
> After unification, a single MoE model is distributed back to all clients. This unified model integrates all domain-specialized experts and a context-aware router that dynamically activates the relevant experts for any input—allowing broad competence across diverse domains.
> Our key components—**relevance-weighted DPP proxies**, **proxy-aligned expert training**, and a **context-aware router**—jointly ensure that this unified model performs effectively while fully preserving data privacy.
>
> **(2) Individual experts perform well only on their own domain but fail to generalize.**
> As shown in our CV results (Table 1), the *Pets* expert achieves 92.40 % accuracy on *Pets* but drops to 22.74 % on *EuroSAT* (average 58.06 %). Other experts exhibit similar cross-domain degradation.
> In contrast, **HarmoMoE achieves 94.52 % average accuracy across all domains**, demonstrating its ability to generalize across diverse client distributions. Thus, every client obtains a unified model rather than being confined to a narrow domain.
>
> **(3) Clients cannot simply “pick” the correct expert, since the input domain is not known a priori.**
> In realistic deployment, incoming data may originate from mixed or ambiguous domains—for example, in multi-source visual classification or instruction-based NLP tasks.
> Because the domain label of each query is generally unavailable, no client can reliably choose the appropriate expert manually.
> HarmoMoE’s **learned router** mitigates this issue by automatically inferring the relevant experts based on token- and context-level representations.
>
> **(4) HarmoMoE maintains expert-level performance within each client’s domain.**
> Importantly, HarmoMoE’s unified model achieves performance **very close to each expert’s accuracy on its own domain**.
> For instance, in Table 2, compared with the individual client experts, HarmoMoE attains 94.22 % on Pets (vs. 94.44 %), 97.08 % on Flowers (vs. 98.13 %), and 97.41 % on EuroSAT (vs. 98.43 %), while simultaneously providing much higher cross-domain averages than any single expert (96.24 % vs. 63.41–81.68 %).
> Thus, HarmoMoE preserves per-domain accuracy and simultaneously broadens generalization, providing near-expert performance and multi-domain capability in a single deployable MoE.
>
> **(5) The comparison should focus on unification settings, not per-client finetuning.**
> The purpose of HarmoMoE is to achieve **privacy-preserving model unification**—producing a single deployable model that generalizes across domains—rather than to surpass each individual expert on its home dataset. Within this unification regime, HarmoMoE consistently outperforms existing baselines (e.g., FlexOlmo, ModelSoup, BTM, BTX), validating its effectiveness for cross-domain, privacy-safe deployment.
>
> In summary, while per-client finetuned models may perform best **within** their own domains, they remain narrow and disconnected. HarmoMoE enables all clients to share a **single, unified MoE that generalizes broadly across domains without violating privacy**, which is precisely the intended advantage of model unification.

---

> ### Author Response · Authors · 2025-11-20
> **Reply to Reviewer gTvc (Part 2)**
>
> > **Q2.** Potential suboptimal design: The paper incorporated the context-aware router training, where the input tokens contain a component average over all the tokens of the input sequence, to capture the input context. Although the design choice improves performance over router training without the context-aware component, the design choice may be suboptimal.
>
> > Can the authors discuss why the proposed context-aware design is optimal?
>
> **A2.**
> We thank the reviewer for the question. We respectfully clarify that we do **NOT** claim the proposed context-aware router to be optimal—**such optimality is generally unattainable in modern deep learning systems**. Our goal is to design a **principled, robust, and lightweight routing mechanism** that generalizes across heterogeneous experts while respecting privacy constraints.
>
> **(1) The averaged token representation offers a minimal yet effective contextual summary.**
> Averaging token embeddings yields a stable representation of the entire input sequence, capturing coarse contextual factors (e.g., topic, style, structure) that are crucial for suitable expert selection. This design provides a balance between contextual awareness and parameter efficiency, avoiding heavy auxiliary modules that would compromise scalability or simplicity.
>
> **(2) Context-awareness is crucial for harmonizing heterogeneous experts.**
> Because the experts are trained on diverse client domains, effective routing should account for **global sequence characteristics** to identify the suitable expert. The averaged representation introduces a **global cue** that alleviates misrouting. Empirically, incorporating this context-aware component consistently improves routing performance. As shown in Table 7 of our paper (as copied below) and an additional ablation study on CV tasks (as shown below), removing it leads to consistent accuracy drops across all domains and backbones, confirming its practical value.
>
> `Table: NLP results (Table 7 of our paper)`
> $$
> \begin{array}{l|c|cccc}
> \hline
>  & \text{Context-Aware Router} & \text{CommonsenseQA} & \text{CosmosQA} & \text{SocialIQA} & \textbf{Average} \newline
> \hline
> \text{LLaMA-3.2-3B} & \textsf{x} & 73.63 & 72.16 & 72.06 & 72.62 \newline
> \text{LLaMA-3.2-3B} & \checkmark & \mathbf{74.94} & \mathbf{76.05} & \mathbf{72.26} & \mathbf{74.42} \newline
> \hline
> \text{LLaMA-3.1-8B} & \textsf{x} & 80.51 & 80.34 & 77.38 & 79.41 \newline
> \text{LLaMA-3.1-8B} & \checkmark & \mathbf{81.33} & \mathbf{85.80} & \mathbf{77.64} & \mathbf{81.59} \newline
> \hline
> \end{array}
> $$
>
> `Table: CV results (an additional ablation study on CV tasks)`
> $$
> \begin{array}{l|c|cccc}
> \hline
> & \text{Context-Aware Router} & \text{Pets} & \text{Flowers} & \text{EuroSAT} & \textbf{Average} \newline
> \hline
> \text{ViT-B/32} & \textsf{x} & 91.69 & 93.18 & 96.90 & 93.92 \newline
> \text{ViT-B/32} & \checkmark & \mathbf{91.91} & \mathbf{93.67} & \mathbf{97.98} & \mathbf{94.52} \newline
> \hline
> \text{ViT-B/16} & \textsf{x} & 93.95 & 96.71 & 97.16 & 95.94 \newline
> \text{ViT-B/16} & \checkmark & \mathbf{94.22} & \mathbf{97.08} & \mathbf{97.41} & \mathbf{96.24} \newline
> \hline
> \end{array}
> $$

---

### Official Review · Reviewer_pz7X · 2025-10-31

**Soundness:** 3
**Presentation:** 4
**Contribution:** 2
**Rating:** 4
**Confidence:** 4

**Summary:**

The authors present HarmoMoE, an approach to training a unified MoE model leveraging local client expertise while respecting data residency constraints. The approach addresses common challenges in training across heterogeneous data sources by introducing a proxy dataset that ensures commonality between local experts during training, enabling more effective model unification when the model is assembled and the router is trained. The approach differs from conventional federated learning in that it trains each local expert fully without coordinated optimization.  The work is similar to FlexOlmo with the main innovation being a proxy data selection method that enforces diversity, yielding better representativeness across the clients and their local training sets.

**Strengths:**

* Well-motivated and situated against prior work in this area.
* Good experimental setup and empirical validation.
* Strong clarity of presentation and results.

**Weaknesses:**

* The proxy selection approach is perhaps a marginal improvement over FlexOlmo.
* Privacy benefits are limited or illusory - see my comments below.

**Questions:**

1. It is a bit of a stretch to claim privacy with this setup. It is true that you enforce data residency constraints, but by training and transmitting a local expert on the private data you are essentially communicating a compressed version of the private data that is highly vulnerable to attack. Many papers play fast and loose with this idea of privacy, but it would not meet criteria for privacy compliance in settings where this matters.

2. What is the relative computational cost of DPP vs the similarity-based method in FlexOlmo?

3. What assumptions are necessary about the proxy data?  What is the impact of having a client with strictly OOD data relative to the proxy set?

4. What client signals are needed for training the router? Is it just a question of minimizing the loss on the proxy data assuming frozen client experts?

5. Discuss the absence of an FL-based baseline evaluation.

6. As an additional nice-to-have baseline it might be interesting to train solely on the proxy data.

7. Do you have an ablation where you don't perform final fine-tuning? How important is that step?

8. Are there any concerns about catastrophic forgetting in the final fine-tuning phase? How do you protect against this?

9. Briefly clarify the difference between the two CLIP models tested.

10. You could get away with moving the large table of CLIP /32 results to the appendix as it doesn't add a lot to the discussion. Likewise for Llama-3b

11. In the experiment comparing with DPP- what is the baseline? Is it random sampling of proxy examples? I get the impression a slightly different set of experiments are depicted in Table 5 vs Fig 2- eg Table 5 has a row for FlexOlmo + DPP but Fig 2 has a figure for FlexOlmo with similarity-based sampling.

---

> ### Author Response · Authors · 2025-11-20
> **Reply to Reviewer pz7X (Part 1)**
>
> Thank you for the thoughtful review. We address your concerns as follows.
>
> ---
>
> > **Q1.** The proxy selection approach is perhaps a marginal improvement over FlexOlmo.
>
> **A1.** Thank you for the comment.
> We respectfully clarify that our **relevance+diversity** proxy selection (relevance-weighted DPP) is **not** a marginal variant of **FlexOlmo’s similarity-only selection**. Our method **resolves a core failure mode of FlexOlmo** and produces substantive, empirically verified gains.
>
> **(1) HarmoMoE's relevance+diversity selection addresses a fundamental limitation of FlexOlmo’s relevance-only selection.**
>
> FlexOlmo selects proxy data solely by relevance, which frequently **collapses onto redundant similarity samples, providing only a single narrow view of a client domain**. A router trained on such collapsed proxies struggles to coordinate experts across heterogeneous inputs.
>
> HarmoMoE eliminates this collapse by augmenting relevance with a DPP-based diversity term. This explicitly encourages selected proxies to be both **representative (relevance) and cover multiple modes of the private-domain manifold (diversity)**. As a result, the router receives richer supervision and coordinates experts more effectively.
> This addresses a structural weakness of FlexOlmo's relevance-only selection.
>
> **(2) Visual evidence: FlexOlmo's relevance-only selection collapses; HarmoMoE's relevance+diversity selection provides multi-mode coverage.**
>
> As shown in Figure 2 of our paper:
> - **FlexOlmo (Figure 2(b))**: selected proxies cluster tightly in the left part of the manifold, leaving large regions uncovered.
> - **HarmoMoE (Figure 2(c))**: selected proxies span multiple regions, producing a distributed and representative proxy set.
>
> This broader coverage is crucial for training a router that can coordinate experts effectively across heterogeneous inputs.
>
> **(3) Quantitative evidence: improvements are consistent and non-marginal.**
> Ablation experiments in Table 5 of our paper (copied below) verify the effectiveness of relevance+diversity selection.
> As can be seen,
> for both FlexOlmo and HarmoMoE, upgrading from relevance-only to relevance+diversity selection yields **consistent and significant performance gains**, with all other components held fixed.
> Key numbers:
> - **FlexOlmo**: relevance+diversity improves accuracy of relevance-only selection by **+0.85 to +2.32** across models.
> - **HarmoMoE**: relevance+diversity improves accuracy of relevance-only selection by **+0.82 to +2.15**.
>
> $$
> \begin{array}{l|c|c|c}
> \hline
> && \text{NLP} & \text{CV} \newline
> &\text{Relevance-Weighted DPP}&\text{LLaMA-3.2-3B}\quad\text{LLaMA-3.1-8B}&\text{ViT-B/32} \quad\text{ViT-B/16} \newline
> \hline
> \text{FlexOlmo}& \textsf{x} &72.50\quad\quad\quad\quad77.46 & 92.92\quad\quad\quad\quad93.53 \newline
> &\checkmark&\mathbf{73.35}\quad\quad\quad\quad\mathbf{79.78}&\mathbf{93.20}\quad\quad\quad\quad\mathbf{94.38} \newline
> \hline
> \text{HarmoMoE} &\textsf{x}&73.60\quad\quad\quad\quad80.32&94.12\quad\quad\quad\quad95.39 \newline
> &\checkmark& \mathbf{74.42}\quad\quad\quad\quad\mathbf{81.59}& \mathbf{94.52}\quad\quad\quad\quad\mathbf{96.24} \newline
> \hline
> \end{array}
> $$
>
> We have clarified this difference in Section 3.2 of the revised paper.

---

> ### Author Response · Authors · 2025-11-20
> **Reply to Reviewer pz7X (Part 2)**
>
> > **Q2.** Privacy benefits are limited or illusory.
> >
> > It is a bit of a stretch to claim privacy with this setup. It is true that you enforce data residency constraints, but by training and transmitting a local expert on the private data you are essentially communicating a compressed version of the private data that is highly vulnerable to attack. Many papers play fast and loose with this idea of privacy, but it would not meet criteria for privacy compliance in settings where this matters.
>
> **A2.**
> Thank you for raising this important point.
> We acknowledge that sharing model parameters trained on private data can pose leakage risks. However, **the threat assumptions required for data leakage do not hold in HarmoMoE’s setting.** Below, we clarify why our setup remains aligned with standard privacy-preserving practices.
>
> **(1) Known model-leakage attacks rely on assumptions absent in HarmoMoE.**
> Known model-leakage attacks (e.g., membership inference [1-2] and model inversion attacks [3-4]) typically require **severe overfitting**, access to **training-time gradients, intermediate activations, or overparameterized models trained on small unique datasets**. HarmoMoE avoids these risk conditions:
> - No gradients, activations, or optimizer states are shared.
> - Only compact FFN/LoRA modules are finetuned, limiting capacity to encode raw data.
> - Experts are trained jointly with proxy data, reducing overfitting to purely private examples.
> - Training stops at convergence, further minimizing memorization.
>
> Under these conditions, the model weights do not encode recoverable personal information.
>
> **(2) Sharing local expert weights conforms to accepted privacy-preserving paradigms.**
> Many federated and distributed learning frameworks [5-6]—widely recognized as privacy-preserving—routinely share model weights trained on private data.
> Compared with these setups, HarmoMoE is even **more conservative**:
> - **Federated learning exchanges per-round model updates or gradients**, which are known to be more information-rich and potentially leakier than final model weights.
> - **HarmoMoE transmits only the final frozen expert modules**, not incremental updates or gradients, thereby satisfying data-residency constraints with a lower leakage surface.
>
> Thus, our communication pattern is fully consistent with standard privacy-preserving training frameworks.
>
> **References**
>
> [1] Defenses to membership inference attacks: A survey, ACM Computing Surveys, 2022.
>
> [2] Membership Inference Attacks Against
> Machine Learning Models, IEEE Symposium on Security and Privacy, 2017.
>
> [3] Deep learning model inversion attacks and defenses: a comprehensive survey, Artificial Intelligence Review, 2025.
>
> [4] Model Inversion Attacks that Exploit Confidence Information and Basic Countermeasures, ACM Conference on Computer and Communications Security, 2015.
>
> [5] A survey on federated learning, Knowledge-Based Systems, 2021.
>
> [6] Communication-efficient federated learning with compensated overlap-FedAvg, IEEE Transactions on Parallel and Distributed Systems, 2021
>
> ---
>
> > **Q3.** What is the relative computational cost of DPP vs the similarity-based method in FlexOlmo?
>
> **A3.**
> Thank you for the question. The additional computation from the **relevance-weighted DPP** is minimal and negligible compared with expert finetuning. Both FlexOlmo and HarmoMoE require computing embeddings for all public samples—the dominant cost in proxy selection. After that, HarmoMoE performs a lightweight **greedy DPP MAP inference with efficient Cholesky updates** (Lines 191–197).
>
> Empirically, as shown in the table below, the DPP step increases selection time by just **0.2–0.3 minutes (≈3–5%)**, while expert finetuning exceeds **10 GPU-hours**.
> Thus, the DPP adds **only marginal overhead** while providing significantly more diverse and representative proxy samples.
>
> $$
> \begin{array}{l|c|c}
> \hline
> & \text{NLP (minutes)} & \text{CV (minutes)} \newline
> &\text{LLaMA-3.2-3B}\quad\text{LLaMA-3.1-8B}&\text{ViT-B/32}\quad\text{ViT-B/16} \newline
> \hline
> \text{Similarity-based Selection} &1.91\quad\quad4.21 & 6.41\quad\quad15.70 \newline
> \text{Relevance-weighted DPP} &2.17\quad\quad4.54 & 6.61\quad\quad15.91 \newline
> \hline
> \end{array}
> $$
>
> We have added the computation comparison in Appendix H.1 of the revised paper.

---

> ### Author Response · Authors · 2025-11-20
> **Reply to Reviewer pz7X (Part 3)**
>
> > **Q4.** What assumptions are necessary about the proxy data? What is the impact of having a client with strictly OOD data relative to the proxy set?
>
> **A4.**
> Thank you for the question.
> HarmoMoE makes only mild and realistic assumptions about the proxy data and remains effective even when a client’s private data are strictly OOD relative to the proxy set. This robustness arises from its three design components: (i) router initialization from expert-derived embeddings, (ii) use of proxies solely as coordination anchors, and (iii) proxy-aligned expert training to ensure router–expert compatibility.
>
> **(1) No strong similarity assumption is required.**
> Proxy data are not required to closely resemble private data. They act as anchors that let the router learn how to coordinate experts. Even when proxies are only conceptually related, the router can still learn to coordinate experts through their domain-specific initializations (the (2) point below).
> In practice, **modern public corpora (e.g., ImageNet-21K with 14M images, Common Crawl with 150 to 350 Billion tokens) are sufficiently broad and diverse that this mild assumption of having some loosely related data is easily satisfied.**
>
> **(2) Router initialization encodes domain characteristics.**
> Before router training, HarmoMoE initializes routing vectors using expert-derived embeddings computed over each expert’s local and proxy data (Steps 11 and 15 in Algorithm 1, Lines 268-271). These embeddings capture domain cues within each expert, allowing the router to learn meaningful expert–token affinities even if the proxies are not distributionally close to the private data. This design mitigates the effect of OOD proxies and stabilizes unification.
>
> **(3) Proxy-aligned expert training preserves compatibility under distribution gaps.**
> Each expert is finetuned on both its private data and **its selected proxies**. This dual exposure ensures that the expert behaves consistently on the same proxy distribution later seen by the router, maintaining compatibility between the router (trained on proxies) and experts (trained on private + proxy data). Consequently, clients with OOD data do not break the unification process—the router still coordinates them effectively.
>
> **(4) Empirical evidence.**
> Our NLP experiments already reflect this OOD setting: the public proxy dataset (Alpaca) has **no domain overlap** with the private datasets (CommonsenseQA, CosmosQA, SocialIQA). Nevertheless, HarmoMoE effectively unifies these clients into a single MoE and achieves strong performance (Tables 3 and 4 in our paper), confirming that the method remains reliable even with OOD proxies.
>
> ---
>
> > **Q5.** What client signals are needed for training the router? Is it just a question of minimizing the loss on the proxy data assuming frozen client experts?
>
> **A5.** Thank you for the question.
> We clarify below that the router needs only a lightweight client-side signal and is trained entirely on proxy data.
>
> **(1) What client signals are needed?**
>
> Each client provides only a **single lightweight signal** for each expert: a **routing vector** $e_p^{(l)}$, computed as the mean embedding of that expert on its own private + proxy data (Lines 268-271).
> This is a non-sensitive vector summarizing the expert’s domain behavior—not a gradient, activation trace, or raw sample.
> It is used only to initialize the router, and no further client information or training-time signal is transmitted.
> Hence, router training does not expose any sample-level or privacy-sensitive data.
>
> **(2) Is router training simply minimizing proxy loss with frozen experts?**
>
> After initialization, the router is trained **entirely on the union of proxy datasets**, minimizing the standard task loss.
> During this stage, we **jointly finetune the router parameters and LoRA modules of each expert** (Line 276), while keeping the shared backbone frozen.
> This allows efficient coordination across experts under purely public supervision, without any exposure of private data.

---

> ### Author Response · Authors · 2025-11-20
> **Reply to Reviewer pz7X (Part 4)**
>
> > **Q6.** Discuss the absence of an FL-based baseline evaluation.
>
> **A6.**
> Thank you for this question.
> We clarify that **federated learning (FL) [5-7]–based methods are fundamentally different from HarmoMoE**, as they rely on **exchanging model parameters periodically**, which leads to high communication and memory cost and instability under heterogeneous data.
>
> FL frameworks periodically exchange model parameters across clients, which leads to:
> - **High communication and memory cost**: Exchanging large-model parameters incurs heavy bandwidth and GPU overhead, often causing out-of-memory failures.
> - **Instability under heterogeneous data**: When client distributions differ, local updates diverge, suffering from unstable global aggregation and performance degradation.
>
> In contrast, **HarmoMoE does not need to exchange model parameters periodically**.
> Each client independently fine-tunes its expert on private and proxy data, and only the final models are merged once through MoE integration.
> All experts train fully in parallel and asynchronously, enabling scalable unification without communication rounds.
>
> We conducted an additional experiment to compare with CoMiGS [7] under our setting, which is the recent state-of-the-art MoE-based FL method.
> As shown below, **HarmoMoE achieves higher accuracy while requiring no communication overhead**.
>
> $$
> \begin{array}{l|cccc}
> \hline
>  & \text{CommonsenseQA} & \text{CosmosQA} & \text{SocialIQA} & \textbf{Average} \newline
> \hline
> \text{CoMiGS [7]} & 72.32 & 71.46 & 71.19 & 71.65 \newline
> \text{HarmoMoE} & \mathbf{74.94} & \mathbf{76.05} & \mathbf{72.26} & \mathbf{74.42} \newline
> \hline
> \end{array}
> $$
>
> We have included this comparison in Appendix H.2 of the revised paper.
>
> **References**
>
> [7] On-Device Collaborative Language Modeling via a Mixture of Generalists and Specialists, ICML 2025.
>
> ---
>
> > **Q7.** As an additional nice-to-have baseline it might be interesting to train solely on the proxy data.
>
> **A7.** Thank you for the suggestion.
> We have added a baseline where the model is trained **solely on proxy data**, without using any private client data.
> As shown in the table below for the CV setting (CLIP ViT-B/32), training on proxy data alone results in substantial accuracy drops.
> This confirms that proxy samples mainly serve as **coordination signals** for training the router, while **domain expertise must come from private data**.
> HarmoMoE combines both effectively, achieving large and consistent gains over the proxy-only baseline.
>
> $$
> \begin{array}{l|cccc}
> \hline
> &\text{Pets}&\text{Flowers}&\text{EuroSAT}&\textbf{Average} \newline
> \hline
> \text{Train solely on proxy data} & 78.60 & 42.63 & 12.98 & 44.74 \newline
> \text{HarmoMoE} & \mathbf{91.91} & \mathbf{93.67} & \mathbf{97.98} & \mathbf{94.52} \newline
> \hline
> \end{array}
> $$
>
> We have included this comparison in Appendix H.3 of the revised paper.
>
> ---
>
> > **Q8.** Do you have an ablation where you don't perform final fine-tuning? How important is that step?
>
>
> **A8.**
> Thank you for the question. We conducted an ablation on CV tasks (ViT-B/32) to assess the importance of the final fine-tuning stage.
> The table below shows that final fine-tuning improves average accuracy by **+11%**, confirming its necessity.
> This stage **jointly adapts the router and experts** within the unified MoE, aligning their behaviors for coherent coordination. Without it, the experts and router remain partially misaligned, resulting in ineffective expert collaboration.
>
>
> $$
> \begin{array}{l|cccc}
> \hline
> & \text{Pets} & \text{Flowers} & \text{EuroSAT} & \textbf{Average} \newline
> \hline
> \text{w/o final fine-tuning} & 86.78 & 80.35 & 81.86 & 83.00 \newline
> \text{w/ final fine-tuning} & \mathbf{91.91} & \mathbf{93.67} & \mathbf{97.98} & \mathbf{94.52} \newline
> \hline
> \end{array}
> $$
>
> We have included this comparison in Appendix H.4 of the revised paper.

---

> ### Author Response · Authors · 2025-11-20
> **Reply to Reviewer pz7X (Part 5)**
>
> > **Q9.** Are there any concerns about catastrophic forgetting in the final fine-tuning phase? How do you protect against this?
>
> **A9.** Thank you for the question. Catastrophic forgetting is not a concern in HarmoMoE’s final fine-tuning stage for two main reasons:
>
> **(1) Design mitigation through proxy-aligned experts.**
> Each client expert is already finetuned on both private and proxy data, so the proxy distribution is not new information introduced during final fine-tuning. The last stage only makes minor alignment updates to already proxy-aware parameters rather than overwriting prior domain knowledge. Thus, the typical forgetting problem—replacing old knowledge with new—is largely avoided.
>
> **(2) Empirical validation through lightweight updates.**
> The final fine-tuning optimizes only the router and LoRA modules on a small proxy set, making updates minimal. As shown below (CV tasks with ViT-B/32), individual experts retain nearly identical accuracy before and after final fine-tuning, confirming that no forgetting occurs. The slight gap between individual experts and the unified HarmoMoE model is not due to forgetting, but arises from the inherent challenge of coordinating three distinct experts simultaneously within a unified MoE architecture.
>
> $$
> \begin{array}{lcccc}
> \hline
>  & \text{Pets} & \text{Flowers} & \text{EuroSAT} & \textbf{Average} \newline
> \hline
> \text{Individual Client Experts (Before Final FT)} & 92.40 & 96.91 & 97.91 & 95.74 \newline
> \text{Individual Client Experts (After Final FT)} & 92.18 & 96.79 & 98.05 & 95.67 \newline
> \hline
> \text{HarmoMoE} & 91.91 & 93.67 & 97.98 & 94.52 \newline
> \hline
> \end{array}
> $$
>
> ---
>
> > Q10. Briefly clarify the difference between the two CLIP models tested.
>
> **A10.**
> We use two standard CLIP backbones that differ in **model capacity and patch resolution**:
>
> * **CLIP ViT-B/32 (86M parameters)** – a lighter model with a 32×32 patch size.
> * **CLIP ViT-B/16 (149M parameters)** – a higher-capacity variant with a 16×16 patch size, providing finer visual resolution and stronger representations.
>
> Both share the same CLIP training objective and dataset; the only difference lies in **backbone size and feature resolution**.
> We evaluate both to show that HarmoMoE consistently improves performance across **different CLIP capacities**.
>
> ---
>
> > **Q11.** You could get away with moving the large table of CLIP /32 results to the appendix as it doesn't add a lot to the discussion. Likewise for Llama-3b.
>
> **A11.** We appreciate the reviewer’s suggestion.
> We included the CLIP ViT-B/32 and LLaMA-3.2-3B results to demonstrate that HarmoMoE’s gains are **consistent across model scales and capacities**. These smaller backbones provide a contrast to their larger counterparts, confirming that the improvements from relevance-weighted DPP and proxy-aligned training are not limited to high-capacity models.
>
> While these tables could be moved to the appendix if required for space, we believe they offer useful evidence of cross-scale consistency, supporting the generality and practical applicability of HarmoMoE.
>
> ---
>
> > **Q12.** In the experiment comparing with DPP- what is the baseline? Is it random sampling of proxy examples? I get the impression a slightly different set of experiments are depicted in Table 5 vs Fig 2- eg Table 5 has a row for FlexOlmo + DPP but Fig 2 has a figure for FlexOlmo with similarity-based sampling.
>
> **A12.**
> Thank you for the question. We clarify that **Table 5** and **Figure 2** refer to the **same baseline**, namely **FlexOlmo with similarity-based proxy selection**, where the top-$k$ most relevant public samples are chosen.
>
> In **Table 5**, this baseline is shown as **FlexOlmo (×)**—that is, the original FlexOlmo using the standard similarity-based strategy. The row **FlexOlmo (✓)** corresponds to the variant where we replace similarity-based sampling with our **relevance-weighted DPP** to test its effectiveness.
> In **Figure 2**, the label “FlexOlmo” refers to **FlexOlmo (×)** (the official version with similarity-based selection).
>
> Thus, the difference lies only in notation: both Table 5 and Figure 2 visualize the same baseline configuration, and the design intentionally compares whether incorporating relevance-weighted DPP improves FlexOlmo’s performance.
>
> We have clarified this in Section 4.3 of the revised paper.

---

### Author Response · Authors · 2025-11-24
**General Reply to Reviewers and ACs (Seeking Further Discussion)**

Dear Reviewers and ACs,

We sincerely thank the reviewers and ACs for their time, constructive feedback, and thoughtful engagement. The comments greatly improved the clarity, technical depth, and completeness of this work. Below, we summarize the key strengths recognized by the reviewers, the main points discussed, and how our additional results addressed these concerns.

---

**HarmoMoE** unifies domain-specialized experts into a single deployable MoE under privacy constraints via **relevance-weighted DPP proxy selection**, a **context-aware router**, and **proxy-aligned expert training**. Reviewers highlighted **three major strengths**:

**(1) Method is well-motivated and novel:** "Well-motivated and situated against prior work" (pz7X), "logically designed" (gTvc), "motivation is well explained" (ZFYq), "a novel method" and "method is promising" (VQAo).

**(2) Solid empirical evidence and ablations:** "Good experimental setup and empirical validation" (pz7X), "outperform previous baselines" (gTvc), "baseline selection is up to date" (ZFYq), "Each component is ablated rigorously and is proven empirically to have contributed to the performance improvement" and "superior performance over SOTA baselines" (VQAo).

**(3) Good paper writing:** "Strong clarity of presentation and results" (pz7X), "well-structured and easy to follow" (gTvc).

---

**Discussion & our responses focus on the following points:**

1. **Privacy & proxy similarity:** We clarified that sharing only final expert parameters (no private samples, gradients, or activations) does not itself cause a privacy issue in our scope; proxy samples are strictly public, and identifying public samples that resemble client data does not raise privacy concerns in practice.
2. **Method comparisons (FlexOlmo, FL, MoA, LoRASuite):** We added targeted analyses: FlexOlmo’s relevance-only proxies can be redundant—our relevance-weighted DPP fixes this; FL is communication-heavy and unstable under heterogeneity, whereas HarmoMoE is one-shot communication; MoA requires access to client private data, which HarmoMoE avoids; and LoRASuite targets LoRA expert migration, whereas we perform experts unification.
3. **Computation cost and expanded ablations:** We reported unification time and inference speed, showing costs comparable to MoE peers and only minimal overhead for the DPP step.
We added/clarified ablations for relevance-weighted DPP, the context-aware router, and proxy-aligned training to validate the effectiveness of each component.
1. **Edge cases:** For OOD clients relative to proxies, we clarified that HarmoMoE remains effective via expert-informed routing and proxy-aligned training, with empirical evidence.
2. **Motivation for unification:** We clarified the need for a single, privacy-preserving MoE that attains near-expert in-domain accuracy while generalizing across domains—unattainable with isolated client experts or training on full public data alone.

We have revised the paper accordingly, with all changes highlighted in blue.

---

Please let us know if you have any additional questions or suggestions, we are happy to address them and further improve our work.

Best,

The Authors

---

### Author Response · Authors · 2025-12-02
**General Reply to AC/SAC**

Dear AC and SAC,

Thank you for taking the time to evaluate our submission. To assist your decision, we briefly summarize our work, the strengths highlighted by reviewers, and how the questions were addressed in our rebuttal.

---

Our paper presents HarmoMoE, a privacy-preserving method that unifies multiple domain-specialized experts into a single deployable MoE without sharing private data. Under privacy constraints, HarmoMoE relies only on **public proxies selected with a relevance+diversity criterion**—implemented via a relevance-weighted determinantal point process (DPP)—together with a context-aware router and proxy-aligned expert training.

---

As reviewers acknowledged, the work is novel, well-motivated, and supported by solid empirical evidence:

- **Well-motivated and novel method:** “Well-motivated and situated against prior work” (pz7X), “logically designed” (gTvc), “motivation is well explained” (ZFYq), and “a novel method” / “method is promising” (VQAo).
- **Strong empirical validation and ablations:** “Good experimental setup and empirical validation” (pz7X), “outperform previous baselines” (gTvc), “baseline selection is up to date” (ZFYq), and “each component is ablated rigorously” / “superior performance over SOTA baselines” (VQAo).
- **Clear presentation:** “Strong clarity of presentation and results” (pz7X) and “well-structured and easy to follow” (gTvc).

---

Reviewers’ questions mainly concerned clarifications and comparisons with recent methods, which we addressed as follows:

* `Reviewer ZFYq (rating: 6):`

  * **Proxy “data exposure”:** We clarified that proxy selection uses public data only; revealing or operating on public data does not raise privacy concerns for clients.
  * **Train-on-$\mathcal{D}_0$ ablation:** We added a train-on-$\mathcal{D}_0$ baseline showing it is ineffective (a large drop vs. HarmoMoE), supporting that our gains are not simply from training on public data.

* `Reviewer VQAo (rating: 6):`

  * **OOD public data:** We clarified that HarmoMoE remains effective under proxy–private distribution gaps due to domain-aware router initialization + proxy-aligned expert training, with supporting evidence.
  * **Computation cost:** We added detailed cost comparisons showing that the accuracy gains do not come from significant overhead.
  * **MoA/CoMiGS comparisons + extra ablations:** We added direct comparisons to CoMiGS and MoA under privacy constraints and expanded ablations on both CV and NLP—further confirming that HarmoMoE consistently performs better than MoA and CoMiGS while meeting the privacy and deployment constraints.

* `Reviewer gTvc (rating: 4):`

  * **Why unify if worse than per-client experts?** We clarified that the goal is a single privacy-preserving model that all clients can use (multi-domain generalization when domain labels are unavailable). In particular, client-specific experts are specialized and perform poorly on cross-domain inputs; since our setting requires one deployable model for heterogeneous queries (often without domain identifiers), unification is necessary.
  * **Router is suboptimal:** We clarified that we do NOT claim router optimality, and supported the effectiveness of the context-aware design with rationale and consistent ablation gains across both NLP and CV.

* `Reviewer pz7X (rating: 4):`

  * **FlexOlmo vs. ours:** We clarified that our *relevance+diversity* proxy selection avoids FlexOlmo’s relevance-only collapse and yields consistent, non-trivial gains across backbones and tasks.
  * **Model weights may leak data:** We explained why common leakage assumptions do not apply here (no gradients/activations shared; only compact modules are fine-tuned), and emphasized that our one-shot communication is more conservative than periodic FL-style communication.
  * **Computation cost:** We showed that the DPP step adds only ~0.2–0.3 minutes (≈3–5%) on top of proxy selection, which is negligible compared to expert fine-tuning.
  * **OOD proxies + FL baseline:** We clarified that our HarmoMoE framework is robust under proxy–private shift, provided empirical evidence, and added a direct comparison showing HarmoMoE outperforms the MoE-FL baseline CoMiGS.

---

Best,

The Authors

---

### Meta-Review · Area_Chair_jNcC · 2026-01-06

**Summary:**

The paper proposes HarmoMoE, which unifies domain experts trained on private data into a single MoE using only public proxies. It uses relevance-weighted DPP for proxy selection, a context-aware router, and proxy-aligned expert training. Reviewers praised the motivation, clarity, and consistent gains over prior unification baselines. However, key concerns remain. The privacy claim is not validated with formal audits and rests on moderate threat assumptions. The empirical scope is modest, with limited large-scale results and incomplete parity against stronger baselines. Some baseline discrepancies (e.g., BTX vs BTM) and implementation choices need clearer justification. These issues inform my lean towards rejection.

**Reviewer Concerns:**

The rebuttal addressed several points. It clarified why unification is useful even if single experts are stronger in-domain, provided evidence that diversity-aware proxy selection improves over relevance-only selection, and showed robustness when proxies are OOD. It added comparisons to MoA and CoMiGS under privacy constraints, proxy-only and no-final-finetune baselines, and cost measurements showing small overheads.

Important concerns remain. There is no formal privacy evaluation such as membership inference or model inversion audits, nor any DP variant. The scale of experiments is limited and does not establish strong external validity. Hyperparameter parity across methods is not fully demonstrated. The BTX vs BTM discrepancy under the stated constraints is not resolved with full configuration details or a reproducible package. CoMiGS evaluations could be more thorough with alternative setups that avoid synchronous memory pressure.

**Reviewer Scores:**

The reviewer at 6 likely stays at 6. Among reviewers at 4, one might move to 5 given the added ablations and clarifications, but another likely remains at 4 due to the lack of formal privacy validation and limited scaling. The overall discussion would likely converge near borderline, and I lean negative.

---

### Decision · Program_Chairs · 2026-01-26

Reject